# Developing meshing workflows in GMSH v4.11 for Geologic Uncertainty Assessment of the High-Temperature Aquifer Thermal Energy Storage

Ali Dashti[1], Jens C. Grimmer[1], Christophe Geuzaine[2], Florian Bauer[3], Thomas Kohl[1]

[1]Institute of Applied Geosciences, Karlsruhe Institute of Technology (KIT), Karlsruhe, Germany
[2]Université de Liège, Institut Montefiore B28, 4000 Liège, Belgium
[3]Institute for Nuclear Waste Disposal, Karlsruhe Institute of Technology (KIT), Hermann-von-Helmholtz-Platz 1, 76344 Eggenstein-Leopoldshafen, Germany

*Correspondence to*: Ali Dashti (Ali.dashti@kit.edu)

**Abstract.** Evaluating uncertainties of geological features on fluid temperature and pressure changes in the reservoir plays a crucial role in the safe and sustainable operation of the High-Temperature Aquifer Thermal Energy Storage (HT-ATES). This study introduces a new automated surface fitting function in the Python API of GMSH (v. 4.11) to model the impacts of arbitrary structural barriers and variations of the reservoir geometries on temperature and pressure in heat storage applications. These structural barriers and geometry variations cannot always be detected by geophysical exploration, but, due to geological complexities, nonetheless may be present. A Python workflow is developed to implement an automated mesh generation routine for varying geological scenarios. This way, complex geological models and their inherent uncertainties are transferred into reservoir simulations. Developed meshing workflow is applied to two case studies: 1) Greater Geneva Basin with the Upper Jurassic ("Malm") limestone reservoir and 2) the 5° eastward tilted DeepStor sandstone reservoir in the Upper Rhine Graben with a uniform thickness of 10 m. In the Greater Geneva Basin example, the top and bottom surfaces of the reservoir are randomly varied ± 10 m and ± 15 m, generating a total variation of up to 25 % from the initially considered 100 m reservoir thickness. The injected heat plume in this limestone reservoir is independent of the reservoir geometry variation, indicating the limited propagation of the induced thermal signal. In the DeepStor reservoir, a vertical sub-seismic fault juxtaposing the permeable sandstone layers against low permeable clay-marl units is added to the base case model. The fault is located in distances varying from 4 m to 118 m of the well to quantify the possible thermohydraulic response within the model. The variation of the distance between the fault and the well resulted in an insignificant change in the thermal recovery (~1.5 %) but up to a ~10.0 % pressure increase for the (shortest) distance of 4 m from the injection well. Modelling the pressure and temperature distribution in the 5° tilted reservoir, with a well placed in the center of the model, reveals that heat tends to accumulate in the updip direction while pressure increases in the downdip direction.

Keywords: HT-ATES, GMSH, Greater Geneva Basin, DeepStor, geological uncertainty, numerical modelling

# 1 Introduction

Aquifer Thermal Energy Storage (ATES) yields the highest storage capacities compared to other energy storage solutions (Fleuchaus et al., 2018). Based on the injection temperature and application, ATES falls into two categories: 1) High-Temperature (>50 °C) Aquifer Thermal Energy Storage (HT-ATES; e.g., Wesselink et al. (2018)), and 2) Low-Temperature
Aquifer Thermal Energy Storage (LT-ATES; e.g., Réveillère et al. (2013)).

Seasonal storage constitutes a low risk in terms of time, budget, and performance (Fleuchaus et al., 2020a). The typically applied "push-pull" concept of HT-ATES facilitates the horizontal transport of large volumes of fluid within an aquifer. Push-pull operation requires a single well for the injection and production (Blöcher et al., 2024). Hence, it is more efficient than the "flow-through" concept, especially in the test phase (Wang et al., 2020). HT-ATES provides a significant advantage
in its reduced site dependence compared to conventional deep geothermal utilizations. It exploits suitable aquifers that can be encountered in the deeper subsurface of major populated urban areas (Schmidt et al., 2018; Mahon et al., 2022). Appropriate reservoir conditions for heat storage are widely distributed in the uppermost 2 km of the continental crust (Bloemendal et al., 2014; Gao et al., 2019; Dinkelman and van Bergen, 2022; Fleuchaus et al., 2020a; Pasquinelli et al., 2020). Suitable reservoirs for thermal energy storage can even exist in thick successions of fractured rocks (e.g., Birdsell and Saar (2020)).
Another advantage of HT-ATES is its minimal surface area requirement, making it an attractive option in densely populated urban areas (Böhm and Lindorfer, 2019).

Development of HT-ATES hinges on appropriate petrophysical properties of the deep aquifer that can be used as a reservoir. Such design requires conceptual geological and numerical models. Most HT-ATES studies describe reservoir geometries as homogeneous kilometer scale box shaped volumes. The sensitivity of these volumes to relevant parameters (e.g., well
configuration, transmissivity, flow rate, and conductivity) has been extensively studied (Stricker et al., 2020; Green et al., 2021; Mindel and Driesner, 2020; Fleuchaus et al., 2020a; Fleuchaus et al., 2020b). The conceptual designs of both, HT- and LT-ATES, typically apply box shaped reservoir simulations while disregarding natural geometries and the impact of geological uncertainties.

Establishing HT-ATES in previously exploited oil fields leverages the data and experiences gained from past exploration and
production activities. Some depleted hydrocarbon reservoirs are re-used for natural gas storage to meet increased demand during the winter season. Compared to $CO_2$ (Li et al., 2006) or $H_2$ (Muhammed et al., 2023) storage, these depleted reservoirs are yet less commonly used for heat. This scarcity of experience necessitates the development of numerical modelling approaches.

Subsurface data inherently encompass varying degrees of uncertainty originating from measurement errors, biased
extrapolations and interpretations, heterogeneities, and simplifications (Caers, 2011; Wellmann and Regenauer-Lieb, 2012; Wellmann et al., 2010; Wellmann and Caumon, 2018). In this study we focus on the impact of structural and geometrical uncertainties on pressure and temperature distribution and their spatio-temporal development in heat storage reservoirs during operation. These uncertainties comprise varying morphologies of the reservoir roof and floor surfaces and vertical

sub-seismic faults that laterally delimit the reservoir, but cannot be predicted from surface measurements. These impacts are often simplified or ignored due to the complexities of re-meshing. Prognostic geological models cannot cope with the uncertainties of the subsurface. Uncertainty analysis highlights the necessity of applying stochastic geological models rather than a deterministic geometrical representation. This study expands the application presented in Dashti et al. (2023) by introducing an automated workflow that generates meshes for complex structural models, enabling the quantification of relevant processes in HT-ATES.

In this study, two potential HT-ATES sites in the vicinity of populated areas are evaluated: 1) the Greater Geneva Basin (GGB) next to Geneva (SW Switzerland) and 2) the designated DeepStor site, located at the campus of Karlsruhe Institute of Technology (KIT; SW Germany). These two locations exhibit significant differences in reservoir geometry, lithology, petrophysical properties, and thicknesses. To assess the impact of structural uncertainties on both the Geneva and DeepStor HT-ATES cases, we designed different scenarios. Quantification of the uncertainty included thickness and geometry variations by adapting a fast, specific meshing workflow. Different scenarios with identical material properties but variating meshes (geologies) are run for each HT-ATES case. The meshing routine generates surfaces from discrete point clouds to create arbitrarily shaped volumes. This automated meshing procedure allows to establish various stochastic numerical models that account for the resolution of the data and even can include an additional vertical fault. Consequently, meshing routines represent the basis for advanced thermohydraulic analyses from arbitrarily inserted faults into the model.

## 2 Uncertainty and Numerical model developments

### 2.1 Greater Geneva Basin

The HT-ATES system proposed for the outskirts of Geneva is situated within the GGB and is designed to store the excess thermal energy, up to 35 GWh, from a nearby power plant (Collignon et al., 2020). For details on the geology of the GGB, refer to Kuhlemann and Kempf (2002). Two formations are recognized as potential heat storage reservoirs: Upper Jurassic Malm limestones and sand rich layers in the Cenozoic Molasse sediments (Chelle-Michou et al., 2017). The geothermal gradient for the GGB is equal to 25-30 K km$^{-1}$ (Rybach, 1992; Chelle-Michou et al., 2017). The 2530 m deep geothermal well (Thonex-01) intersected >900 m thick Malm limestones and marl succession with a bottom hole temperature of 88 °C and low flow rates of <0.5 l s$^{-1}$ (Guglielmetti et al., 2022). The gradient is not very promising for geothermal heat production from the reservoir, but heat storage can efficiently support the higher heat demand during the winter season. The flow rate has also been low due to the reservoir's characteristics in that specific location.

Collignon et al. (2020) conducted a local parametric sensitivity analysis on the Molasse and Malm limestone reservoirs of the HT-ATES. The proposed target Malm limestones consist of patch reefs with high porosities (Chevalier et al., 2010; Rybach, 1992). In their scope study, Collignon et al. (2020) assumed a box shaped reservoir with flat top and bottom surfaces at -1100 m and -1200 m depths, respectively. Our study simulates the pressure and temperature fields in the geometrically variating Malm reservoirs while the material properties are fixed and identical. We investigate the impact of

the geological uncertainty caused by the carbonate reservoir. Such uncertainties typically stem from the exploration of a reservoir structure that can be based on earlier seismic data acquisition (Feng et al., 2021; Faleide et al., 2021). The sources of error comprise data acquisition, preprocessing, stacking, migration, availability of well data for depth calibration, quality of velocity models for time-depth conversion, and ambient noise level (Bond, 2015; Thore et al., 2002).

To perturb the geological model, a randomized error is superimposed on the top and bottom surfaces of the initial box shaped reservoir. This error is introduced randomly due to the lack of any real geologic model. This study follows the work performed on a generic box with flat surfaces by Collignon et al. (2020); consequently, the considered uncertainty also remains generic and random numbers are chosen as the error values to avoid any bias. For the top surface, a range of ± 10 m arbitrary error is imposed on the primary flat plane. For the bottom surface, the range of perturbation is increased to ± 15 m

due to the decrease in the quality of seismic data with depth. The reasoning behind these arbitrary values of 10 m and 15 m, as well as their increase with depth, is elaborated by Lüschen et al. (2011) and Stamm et al. (2019), respectively. The availability of the well data allowed for well-to-seismic tie which increases the accuracy. In the geological model, it is assumed that at intersections of the wells with the top (-1100 m) and bottom (-1200 m) surfaces of the reservoir, the depth value is a certain data. A simplified 2D schematic is presented in Figure 1-a to visualize the process of assigning generic

uncertainty to the depth data of the GGB. As shown in the figure, the base case assumes the simplest geometry and all scenarios must pass through the four certain points.

For the Malm limestone reservoir, a grid of discrete points in x, y, and z coordinates of a 3D space (representing surfaces) is generated. The regular grid consists of 41×26 nodes in x and y directions, respectively with a fixed 20 m distance. The perturbed model is a purely generic example where at each grid point the random error is added to its vertical coordinate similar to the 2D example in Figure 1-a. For the grid points representing the top surface, any value from -10 to +10 has been

generated and added to their initial vertical coordinates, i.e. -1100 m. The same process applied for the bottom surface but with a bigger range of error (-15 to +15). In realistic cases, geological surfaces may be subjected to other sources of uncertainty. For instance, a function could be defined to establish a direct relationship between the error value and the distance from the wells, addressing spatial correlation. However, this approach could lead to generating a reservoir with

concave or convex surfaces, while meshing highly complex surfaces is one of the contributions of this study.

Figure 1-b presents a scenario with two perturbed surfaces of the Malm limestone layer. The irregularity of the reservoir's undulating surfaces is observable in this figure. The entire discretized model includes basement, reservoir and caprock as lower, middle and upper units, respectively.

(a)

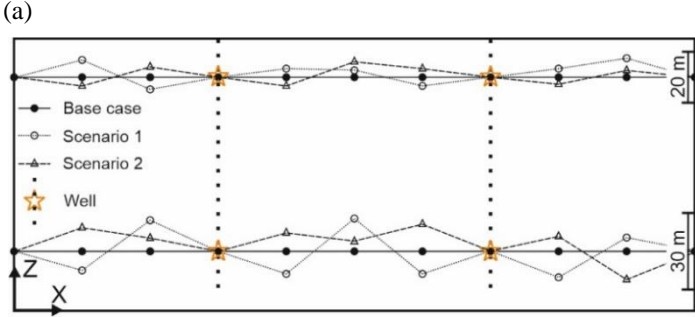

(b)

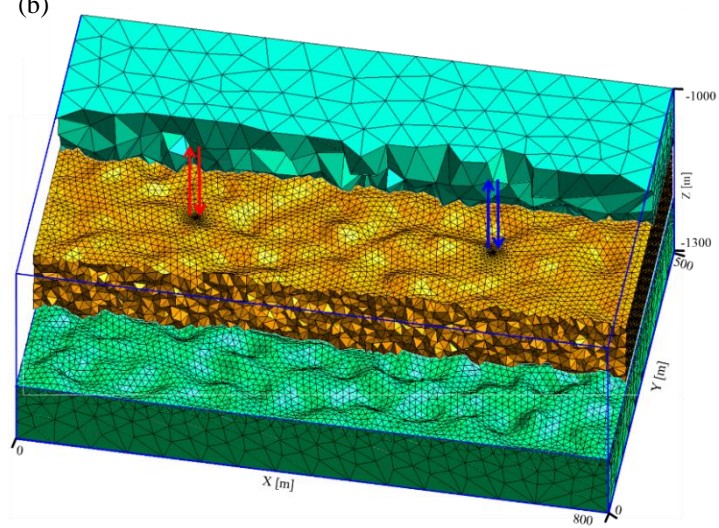


**Figure 1: a) The solid line passing through black dots represents the base case. In each of the three scenarios, the geometry of the reservoir is different but all the lines pass through the orange stars which highlight the contact points of the wells and reservoir. b) The entire discretized model of a perturbed scenario. The reservoir layer in the middle is sandwiched by the basement and caprock units. Red arrows represent the injection and production operations in the hot well whereas the cold well is shown with**
**blue arrows.**

## 2.2 DeepStor

The proposed DeepStor site is located in the Cenozoic sediments of the Upper Rhine Graben (URG) and aims to use an abandoned and depleted oil field for thermal storage in the sand layers of the Oligocene Meletta beds. For details on the geology and stratigraphy of the URG, refer to Grimmer et al. (2017), Dèzes et al. (2004), and Schumacher (2002) and
references therein. Figure 2 highlights the abundance of N-S striking normal faults in the URG that – if suitably oriented in the stress field – facilitate convective fluid flow in fractured Permo-Mesozoic and crystalline basement rocks. Convection in fractured Permo-Mesozoic rocks creates positive thermal anomalies in the Cenozoic graben filling generating locally geothermal gradients of up to 100 K km$^{-1}$ (Agemar et al., 2012; Baillieux et al., 2013; Pribnow and Schellschmidt, 2000).

DeepStor designated HT-ATES aims to utilize the Oligocene Meletta sandstones that were exploited for oil from 1957 to

1986 (Reinhold et al., 2016) in the footwall of the sealing Leopoldshafen fault where oil and some gas accumulated updip
(Wirth, 1962; Böcker et al., 2017).

The DeepStor model in this study encompasses a volume with $1000 \times 1000$ m$^2$ area and 250 m height (see Figure 3-a with the
sand layers of the Meletta beds). Due to the inherent uncertainties, sub-seismic faults characterized by offsets <20 m cannot
be accurately identified using either 3D seismic or well data. These faults can laterally delimit reservoir layers and impact

heat storage potentials and operations (Glubokovskikh et al., 2022). Mathematical models have been developed to
characterize these faults because of their abundance and importance (Gong et al., 2019; Rotevatn and Fossen, 2011; Harris et
al., 2019; Damsleth et al., 1998; Wellmann and Caumon, 2018). While sub-seismic faults are expected to exist, their location
in the subsurface remains largely unknown.

To evaluate the impact of sub-seismic faults on HT-ATES operation, a hypothetical N-S striking fault is introduced in

different parts of the basic geological model. The strike of this vertical fault is parallel with Stutensee and Leopoldshafen
faults (Figure 2). The fault remains as a planar 2D surface due to the lack of any information about the hypothetical sub-
seismic fault. In this study, the uniform 15 m vertical displacement of the fault exceeds the thickness of the reservoir. This
pessimistic assumption enables the prediction of the worst case scenarios for the storage in which a fault completely blocks
the reservoir by juxtaposing it against the impermeable matrix. If the offset is reduced and some contacts between the

reservoir on either side of the fault are permitted, the effect of the fault diminishes. Our modelling results are also applicable
for faults with larger dip-slip displacements.

 The single test well (a hot one) is positioned in the center of the model (Figure 3). This arrangement aligns with real storage
cases where a test well allows for an optimal design. Data from this well is subsequently processed to establish a potential
relationship between measured pressure values and the location of a fault. This study evaluates the impact on reservoir

temperature and pressure through thermohydraulic simulations for 16 fault locations. In total, 17 scenarios are considered in
which the parameterization scheme remains the same but the geology (mesh) varies:

- Fourteen scenarios with a fault varying from 4 m to 112 m distances east of the well
- Two scenarios with a fault in the west of the well at 8 m and 48 m distances
- One fault free base case

The 4 m to 112 m range is chosen to evaluate the effect of the fault on the heat propagation and also examine the possible
impact of the fault distance on the pressure response at the well location. Figure 3-b depicts a scenario with an arbitrary fault
located 98 m in the east of the well.

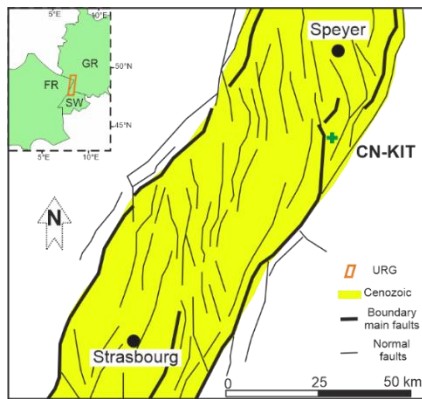

**Figure 2: A tectonic overview of the URG and its surrounding area. The green plus symbol indicates the proposed location for HT-ATES in the north of Karlsruhe. Bold lines mark major faults of the rift boundary fault system. DeepStor site is located between Leopoldshafen (in west) and Stutensee (in east) normal faults (modified from Grimmer et al. (2017)).**

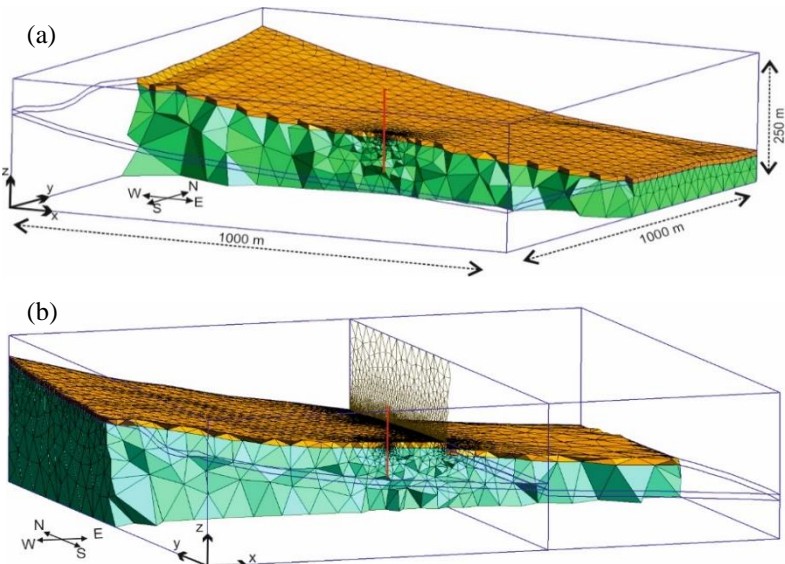

**Figure 3: a) A section across the permeable reservoir layer (orange) and basement (green) of the DeepStor base case. Impermeable clay caprock is not shown to have a better view of the discretized model and morphology of the reservoir. b) A fault is introduced in the model. The dimension of the faulted model remains the same as the base case (1000×1000×250). The fault surface of this example is located 98 m east of the well. In both subplots, the well location is shown via a red line.**

A simplified example in Figure 4 illustrates how the fault embedding is developed for the DeepStor model. Figure 4-a
depicts two surfaces with different colours representing the simplified top and bottom surfaces of the DeepStor reservoir. For a better visualization, surfaces are divided into patches and grid points are labelled with numbers ranging from 1 to 36. In reality, a single surface is generated that fits the grid points of the upper surface (18 black dots) and the same for the lower surface (18 black triangles) (Figure 4-a). The fault displaces the reservoir layer as shown in Figure 4-b. The outline of the

fault in the model is represented by thick red lines passing through points 10, 11, and 12 on the top surface and points 28, 29, and 30 on the bottom surface. The first limitation of the workflow is that can only be placed at existing grid points within the geological model. The workflow is developed to incorporate only N-S striking faults that is its second limitation. Another limitation is the dip angle of the arbitrary fault. The script also simplifies faults to be vertical, neglecting the possibility of inclined faults.

The well in the simplified example indicates the certain depths of the top and bottom surfaces in the model. In the faulted example, the top surface will be divided into two splits: the first split including point numbers from 1 to 12 (left hand side of the fault) and the second one with point numbers 10 to 18 (right hand side of the fault). The left hand side split of the fault does not move and only the right one is displaced downward by the amount of the offset. This approach is used in this example because the well is located within the left hand side split. For each split, an extra set of points are also considered to ensure that the split is properly intersected by the fault plane. In the first split of the top surface, point numbers 13, 14, and 15 are added. One single surface fitting to point numbers from 1 to 15 will be generated for this split. Hashed patches in Figure 4-b show how the extra points are allowing the first split of the top surface to extend toward the fault plane. For the second split of the top surface, points 7, 8, and 9 are additionally included. The second split of the top surface passes through 12 black dots numbered from 7 to 18. This surface generation process is repeated for the bottom surface, whose points are represented by black triangles. Finally, the fault plane will also be generated that intersects each split of the top and bottom surfaces. The extra patches and their corresponding points and lines can be deleted after generating the correct geometry. The explained process allows displacing the grid points of the DeepStor base case or GGB. All the explained steps are implemented and fully elaborated in an example (see Code and data availability section).

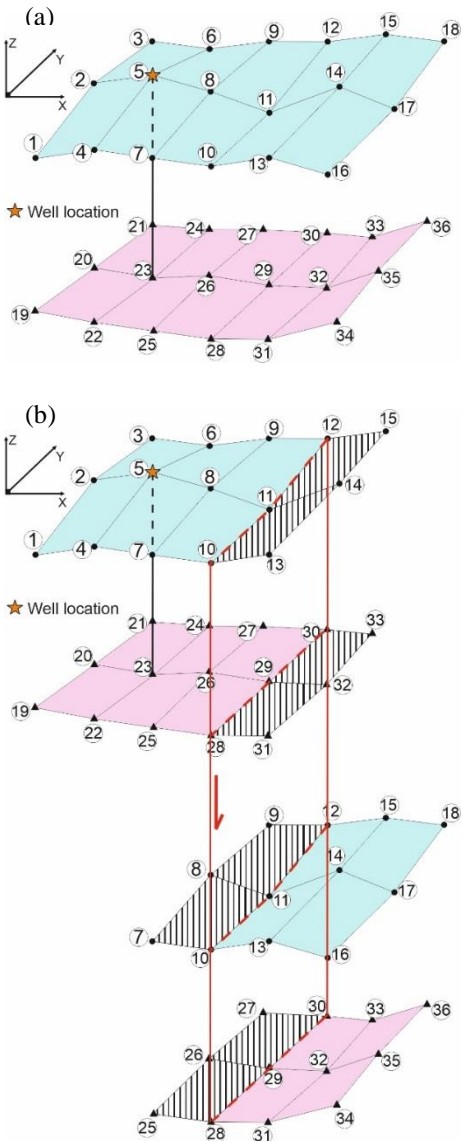

**Figure 4: a) The top and bottom surfaces of the simplified reservoir layer are represented via blue and pink patched surfaces,**
**respectively. Black dots represent the grid points of the top surface while the bottom surface passes through the black triangles.**
**The well location and trajectory are shown via an orange star and a black line, respectively. b) A normal fault with an arbitrary**
**offset is displacing the hanging wall (right hand side splits) downward. Hashed patches are the extra ones added to each split.**

## 2.3 Tool developments based on GMSH

The open source finite element mesh generator GMSH (Geuzaine and Remacle, 2009) is used to generate the required high
quality spatial discretization. GMSH recently gained the ability to create geometrical surfaces passing through arbitrary sets
of points and to combine these surfaces with other geometrical entities (curves, surfaces or volumes) through Boolean

operations thanks to the built-in OpenCASCADE geometry kernel (Open CASCADE Technology). The new features linked to B-Spline surface interpolation and non-manifold meshing are available in the latest stable version of GMSH (v. 4.11). This allows to preserve the geological topology of the layers and enables the generation of high quality, adapted finite

element meshes for complex geometries like modified Malm limestone reservoir surfaces (Figure 1) or tilted Meletta beds (Figure 3). While the model of Dashti et al. (2023) lacks complicated geometries, the recently added functionality of GMSH is tested in this study by implementing complex geometries. The overall workflow for the spatial discretization is based on the following steps that are implemented in fully elaborated scripts using the Python API of GMSH (see Code and data availability section):

• The global outline of the domain of study is defined by adding a single (solid) volume – usually a parallelepiped;

- Geological layers are defined by fitting, through numerical optimization, a B-Spline surface going through each set of grid points defining a geological interface. The grid point cloud can come from any modelling tool and the only requirement is that they should make a regular grid. Simplified schematics like Figure 4 show how the input point cloud can look like. GMSH only requires the x, y, and z values of each point. Default parameters for the B-Spline

degree and the tolerances for the fitting ensure a smooth surface with reasonable local curvature changes;

- Sources and wells (or other zero- or one-dimensional features) are defined as additional points and curves in the model;

- All the geometrical entities are intersected globally in order to produce a conforming boundary representation of the complete model, possibly with non-manifold features (points and curves "embedded" in surfaces and/or volumes);

• Mesh size fields are automatically defined to refine the mesh when approaching the boundaries of the reservoir, as well as when approaching the wells and/or the sources;

The global unstructured mesh is then generated automatically. The mesh is made of tetrahedra inside volumes, triangles on the interfaces, lines on the wells and points on the sources. This mesh is conforming, i.e. the elements are arranged in such a way that if two of them intersect, they do so along a face, an edge or a node, and never otherwise. It is necessary to first

generate the desired number of scenarios for uncertainty analysis and later on one single block of code in Python will yield the same number of meshes.

In the GGB cases and also the base case of the DeepStor, only two surfaces representing the top and bottom of the reservoir are generated in the mesh. In the faulted cases of the DeepStor (Figure 4-b), the grid points making the top and bottom surfaces of the reservoir are discontinuous due to the presence of the fault. Therefore, GMSH should make four different

surfaces to reconstruct the faulted scenarios. As visualized in Figure 4-b, each split is extended to intersect the fault surface, resulting in some additional small patches. These extra parts can be removed in GMSH before meshing. Fully elaborated Jupyter notebooks are provided (see the Code and data availability section) to detail the meshing process for both the DeepStor and GGB cases.

Multi-level mesh refinement is implemented in both models using various functions available in GMSH. In the GGB case,

Distance and Threshold fields enable a gradual mesh size increase from 2 m to 75 m, starting from the wells and extending

towards the model boundaries. Additionally, the mesh size is set to 15 m near the top and bottom surfaces of the reservoir and gradually increases to 75 m. On average, GGB meshes contain approximately 35'000 nodes and 210'000 elements. The average is presented due to the scenario specific variations in the mesh caused by geometrical differences. The fast and automated workflow facilitates the generation of meshes for complex geological models, such as the perturbed GGB

scenarios, within 80 seconds on a Core i7 laptop. Notably, the running time encompasses the entire process from importing data into GMSH to exporting a refined conforming mesh.

DeepStor employs the same refinement strategies but with different mesh sizes. The minimum mesh size is set to 0.5 m near the single well and gradually increases to 125 m. The model also includes a large 2D fault plane. Distance and Threshold fields are introduced for the fault plane, forcing the mesh size to be 3 m near the fault. The DeepStor base case contains

9'026 nodes and 62'317 elements. The mesh is generated in 45 seconds for this fault free case. For the 16 scenarios with the sub-seismic fault, the number of nodes and elements increases to 37'000 and 250'000, respectively. To achieve the specified mesh sizes in both GGB and DeepStor cases, a mesh sensitivity analysis was conducted to ensure the independence of simulation results (temperature and pressure fields) from the mesh size.

## 2.4 Numerical modelling

The open source finite element application TIGER (Thermo-Hydro-Chemical sImulator for Geoscience Research) (Gholami Korzani et al., 2020) is used to simulate the heat storage processes for GGB and DeepStor cases. TIGER is developed on top of the MOOSE (Multiphysics Object Oriented Simulation Environment) framework. As a general purpose PDE environment, the MOOSE framework is fully coupled and encompasses a wide variety of completely implicit solvers (Lindsay et al., 2022; Gaston et al., 2009). It inherits functionalities from PETSc which is a suite of data structures and

routines applied for scalable parallel solution and libMesh that allows for generating and also reading spatial discretization. In our study, the coupled thermal and hydraulic kernels of TIGER are deployed to obtain the evolution of temperature and pressure. To reproduce the results, other MOOSE based applications like GOLEM (Cacace and Jacquey, 2017) or available modules of MOOSE, e.g. Porous Flow (Wilkins et al., 2021), can be used. In TIGER, the mass transport equation (given by mass balance along with the Darcy velocity) is used to simulate the hydraulic behaviour of the system. For heat transport,

TIGER uses the advection-diffusion equation (Gholami Korzani et al., 2020). TIGER simplifies the meshing by enabling a mixed-dimensional problem formulation. Therefore, we considered the wells and faults in the mesh as 1D lines and 2D surfaces, respectively.

Used thermal and petrophysical data for simulation of both cases are directly obtained from the published models. Table 1 contains the values selected for required parameters in our simulations. Considering homogenous petrophysical properties

for patch reefs is highly idealized, but we adhere to the available published data in this instance. Otherwise, a wide range of uncertainty/heterogeneity can be considered for each input parameter. Collignon et al. (2020) used MATLAB Reservoir Simulation Toolbox to simulate the thermohydraulic processes. In this study, simulation results (heat plume propagation and recovery) are compared and benchmarked against their work.

The GGB model includes a doublet system simulated over 10 years. The loading, unloading, and resting phases of the model follow the strategy introduced by Collignon et al. (2020). Each annual cycle comprises four months of loading, two months of rest, four months of unloading, and two months of rest. The loading phase corresponds to the injection of hot water via the hot well when the cold well is in production mode. Temperatures for hot and cold fluid injection are set to be 90 °C and 39 °C, respectively. Both wells have a fixed flow rate of 10 l s$^{-1}$ but in different directions. The MOOSE control system dynamically updates the temperature boundary condition (BC) during the simulation. In the injection phase, the temperature BC is applied to the corresponding nodes in the model, either set to 90 °C or 39 °C. During the production phase, the temperature BC is deactivated. The time stepping for 10 years of simulation is divided into 10 loading, 10 unloading and 20 rest phases. The piecewise linear function of MOOSE is used to increase the time steps in each phase to have a more efficient numerical convergence. During the first cycle (four months of injecting hot fluid into the hot well and producing from the cold well), the time step size increases from one hour to 10 days. Subsequently, the time step size decreases to one hour at the beginning of the rest cycle and gradually increases to 20 days at the end. At the start of the next four month cycle (producing from the hot well and injecting cold fluid into the cold well), the time step size is forced to be one hour and increases to 10 days. For GGB, the simulation runtime is approximately 3 hours on 12 cores of a high performance computing (HPC) cluster with 62 gigabytes of random-access memory (RAM).

Stricker et al. (2020) introduced the properties of the reservoir for DeepStor in a generic model and we used the data of their reference case (Table 1). In our simulations, the geology and consequently the mesh is the major difference to the model of Stricker et al. (2020) while the parameterization scheme remains the same. Rather than the doublet model described by Stricker et al. (2020), a single "push-pull" well is demonstrated in our study. Herein we focus on the thermohydraulic impacts in the near field of a single well in a model with a fault plane. In our meshing procedure, faults (as 2D planes) are integrated only for displacing the 3D elements. They do not have any significance for the MOOSE simulation and can be considered as being only a virtual plane without any physical property because they control the hydraulic behaviour of the model by juxtaposing the reservoir layer against the impermeable matrix. The simulation time is set to 10 years. Hot fluid with a temperature of 140 °C is injected in a six month period, followed by six months of production operation. The MOOSE control system is again applied to switch the temperature BC between injection and production cycles. The flow rate is fixed at 2 l s$^{-1}$ in both the injection and production phases. The time discretization follows the six month cycles and consists of 20 temporal frames for the whole simulation time. Time steps increase from 10 minutes to 10 days in each cycle. Time steps at the start point of each cycle are considered to be shorter in the DeepStor simulations compared to GGB due to the lower thickness of the reservoir and higher complexity of the model. Almost 74'000 degrees of freedom in the faulted scenarios demand an average of 4 hours of computation time on 12 cores of an HPC cluster with 62 gigabytes of RAM. Simulations in the faulted scenarios of the DeepStor are computationally more demanding compared to GGB due to the complexity of the model.

For both the GGB and DeepStor cases, similar approaches are applied for defining boundary and initial conditions. After running a steady state thermohydraulic simulation for each scenario, the results have been applied as the initial condition for

transient simulation of that specific case. In both the steady state and transient simulations, two Dirichlet BCs are also applied for the temperature at the top and bottom surfaces of each model. By introducing a function that represents the temperature gradient, MOOSE allows for assigning the correct temperature values to the model. The depth-dependent temperature function is mentioned in the following:

$$T(z) = T_{surface} + z \times GT, \qquad (1)$$

where z denotes the depth (in m) and GT the geothermal gradient (in K km$^{-1}$). In the case of pressure, one Dirichlet BC is defined on the bottom surface of the model based on the following function for both the steady state and transient simulations (assuming hydrostatic equilibrium):

$$P(z) = (z - WT) \times \rho \times g, \qquad (2)$$

where WT represents the water table depth (in m), $\rho$ is the density (in kg m$^{-3}$) and g is the gravitational acceleration (set as 9.81 m s$^{-2}$).

Neither temperature nor pressure BCs are set on the side faces, hence they follow the gradient. No flow BCs are considered for side faces of the models. The sizes of the models are also big enough to avoid any interaction between the pressure and temperature values of the boundaries and injection/production operation.

**Table 1: Parameters selected as inputs for the numerical simulations of two case studies.**

| Parameter | | | Case studies | |
|---|---|---|---|---|
| | | | GGB (Collignon et al., 2020) | DeepStor (Stricker et al., 2020) |
| Reservoir | Thickness [m] | | ~100 | 10 |
| | Permeability [m$^2$] | | $9.8 \times 10^{-15}$ | $6.6 \times 10^{-14}$ |
| | Porosity [-] | | 0.15 | 0.15 |
| | Thermal conductivity [W m$^{-1}$ K$^{-1}$] | | 1.8 | 2.5 |
| Caprock and basement | Thickness [m] | | ~100 | ~100 |
| | Permeability [m$^2$] | | $9.8 \times 10^{-19}$ | $10^{-18}$ |
| | Porosity [-] | | 0.05 | 0.15 |
| | Thermal conductivity [W m$^{-1}$ K$^{-1}$] | | 1.4 | 1.4 |
| Flow rate [l s$^{-1}$] | | | 10 | 2 |
| Geothermal gradient [K km$^{-1}$] | | | 26 | 50 |
| Water table [m] | | | 10 | 10 |

## 3 Results

### 3.1 GGB

The upper and lower contacts of the reservoir are perturbed to investigate their possible effect/s on the heat and pressure distributions in the HT-ATES. The heat recovery of the system has remained unaffected due to its dependence on local temperature values. Despite changing the geometry of the reservoir, propagation of the heat also appears the same for the three presented scenarios of GGB in Figure 5. Temperature values of the highlighted traces in Figure 5 are extracted to visualize the heat plume propagation. The uppermost scenario in Figure 5 is the base case (a box shaped reservoir with flat

planes) while the two next ones are named as scenarios 1 and 2 in Figure 6. Even after 10 years the heat is still locally propagated (~40 m) around the hot well for the base case and the other two perturbed scenarios (Figure 6). The overlap of all three curves confirms the independence of the temperature field from the introduced geometrical perturbation of the thick reservoir layer.

In addition to the three scenarios presented in the study, eight other geometries are meshed and simulated. The results

indicate that the storage capacity (temperature production) remains consistent across the simulated scenarios. For further analysis, 101 different geometries are generated and uploaded (refer to the Code and data availability section).

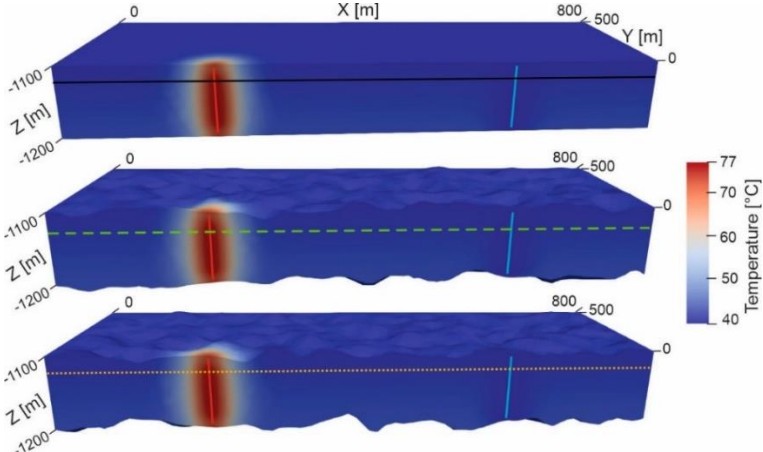

**Figure 5: Heat distribution after 10 years of storage in the Malm limestone reservoir of the GBB. Red and blue lines represent hot and cold wells, respectively. The upper scenario with a uniform box shaped reservoir is considered as the base case while contacts**

**of the reservoir in the middle and lower scenarios are perturbed. Solid black, dashed green and dotted orange traces are used in Figure 6 for plotting the temperature values.**

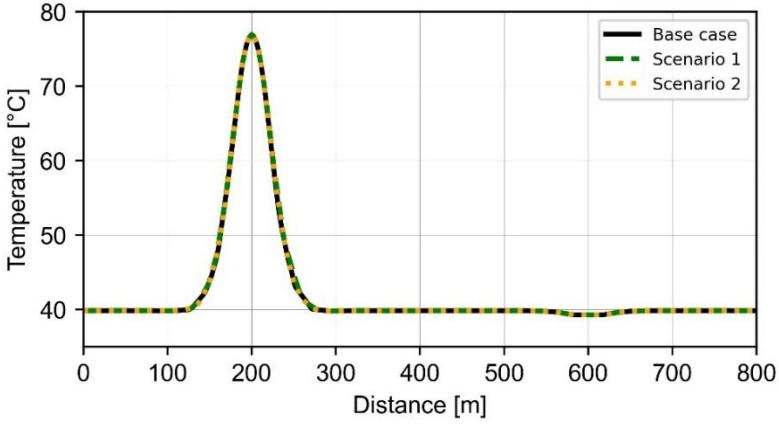

**Figure 6: Temperature distribution curves of the values coming from the base case and two perturbed scenarios after 10 years for the GGB. Hot and cold wells are located at 200 m and 600 m points of the x axis. To find the location of the plotted traces, refer to Figure 5. The extension of the model in x direction (Distance) ranges from 0 m to 800 m.**

### 3.2 DeepStor

Despite incorporating the reservoir's real geology into this study, the recovery and heat plume radius (45 m) of the base case are similar to what is presented by Stricker et al. (2020) for their reference case. The recovery rate is calculated as the ratio between extracted and injected thermal energy at the top of the well's openhole section. Therefore, this parameter only covers the data from one single point of the 3D model and is unable to see the difference between complex and simple reservoir structural models. Figure 7 shows an increase in heat recovery from 67 % to over 82 % between the first and tenth years. The difference between 17 simulated cases is insignificant (~1.5 %). Cases with the highest difference, i.e. extremes, are plotted in Figure 7 to keep the figure readable. The recovery difference between scenarios increases over time, as evidenced by the divergence of the three recovery curves. Despite the negligible difference, the case with a fault located 48 m in the west of the well has the best recovery while the case with a fault in 4 m distance in the east is the worst. For the best recovery, the reason is linked to the total volume of the reservoir and upward movement of the low density hot fluid. The reservoir is tilted and hot fluid moves to the updip direction due to the density effect. Then, a barrier in the updip (west) side of the reservoir can block the movement of the hot fluid and make a more efficient heat storage reservoir. The reason behind the worst recovery is that heat loss happens through the reservoir and matrix contact.

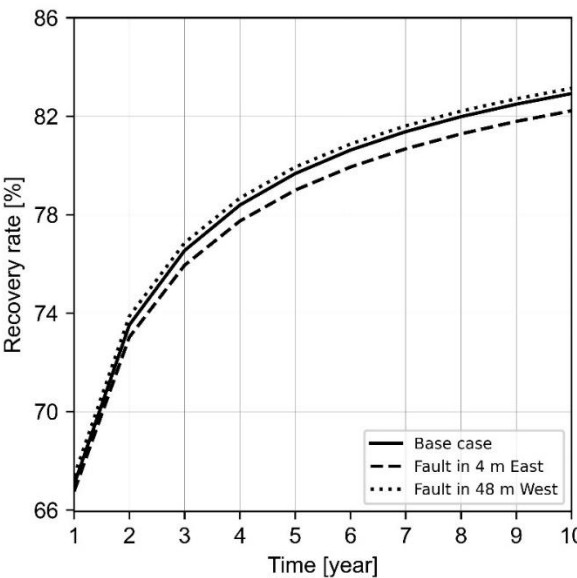


**Figure 7: Heat recovery in three scenarios of the DeepStor model. Only two extremes and the base case are plotted to keep the plot more readable.**

Figure 8 shows the heat accumulation in four distinct simulated scenarios. In the base case (Figure 8-a), the radius and temperature of the heat plume corroborate the results of Stricker et al. (2020). The heat plume extends approximately 45 m in

the x and y directions. The primary distinction is that the heat plume's slope aligns with the tilted reservoir in this instance. The angle between the vertical well and tilted heat plumes in Figure 8 indicates this 5° inclination. The heat plume is most severely affected in the case where the fault is located 4 m in the east of the well (Figure 8-b). When the fault is moved to the edge of the plume (45 m in the east: Figure 8-c), the heat plume appears nearly identical to that of the base case. The resemblance between Figure 8-a and c suggests that the impact of the fault on the heat plume diminishes. The heat plume is

getting slightly warmer when the fault is assumed to be 48 m on the west side of the well (Figure 8-d). Recovery curves also confirmed the higher efficiency of this scenario. After injecting hot water, it flows toward the updip direction of the reservoir due to its lower density. Over a 10 year simulation time, such localization of the reservoir can increase the recovery but in a longer period, these barriers reduce the available storage capacity of the reservoir.

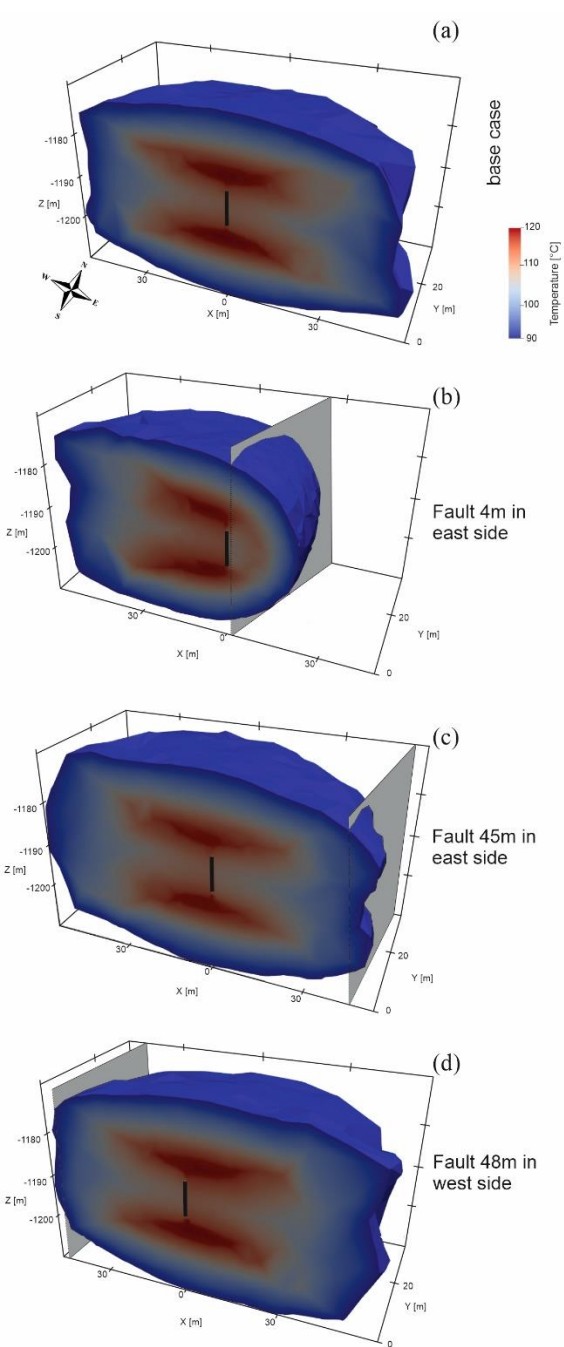

Figure 8: Heat accumulation in four different scenarios of the DeepStor model at the end of the last production cycle. The planned well is shown as a solid black line. Subplots from a to d represent different scenarios including base case, arbitrary fault (shown with a grey surface) in 4 and 45 m in the east of the well and 48 m in the west. The temperature scale is also the same and shown only once in subplot a to avoid repetition.

Figure 9-a and b show a 2D section of the model and the total pressure (hydrostatic plus operation induced pressure) values

across the sand reservoir after the first injection cycle. Ten injection (and production) cycles are included in the simulation and the maximum pressure increase is observed at the end of the first injection cycle. The plotted trace of the pressure curves in Figure 9-b is shown as a dashed line in the cross section view (Figure 9-a). The pressure curves illustrate the data of five cases and the initial condition of the trace passing through the reservoir. The pressure increase of the base case at the well location from the initial condition to the end of the first injection cycle is approximately 10 % (~11.52 MPa to ~12.61 MPa).

The initial condition of the model shows that pressure values are distributed asymmetrically in the reservoir. This distribution confirms the role of the reservoir's inclination on pressure in the model. The eastern part of the reservoir layer is dipping downward and under higher hydrostatic pressure. Therefore, in the majority of the faulted scenarios (14 out of 16), the hypothetical fault is located on the eastern side of the well to present the worst case scenarios and enable a better assessment of the maximum potential pressure increase. Even in the worst case scenario (the fault is 4 m in the east of the

well) the pressure value at the fault is only 7 % higher than the value in the same location of the base case. The total pressure at the fault location of the worst case is 13.1 MPa while in the base case, it is 12.25 MPa. Figure 9-b also suggests a relation between the pressure increase and the location of the fault.

The impact of the fault on the temperature and pressure fields of one case from DeepStor is presented in Figure 10. This figure depicts a small slice from the center of the model, spanning an area of 600 m by 250 m in x and z directions,

respectively. The results demonstrate that the embedded fault effectively creates a barrier very close to the well by introducing a substantial offset. Nevertheless, parts of the injected heat diffused from the reservoir into the matrix, as evident in Figure 10-a.

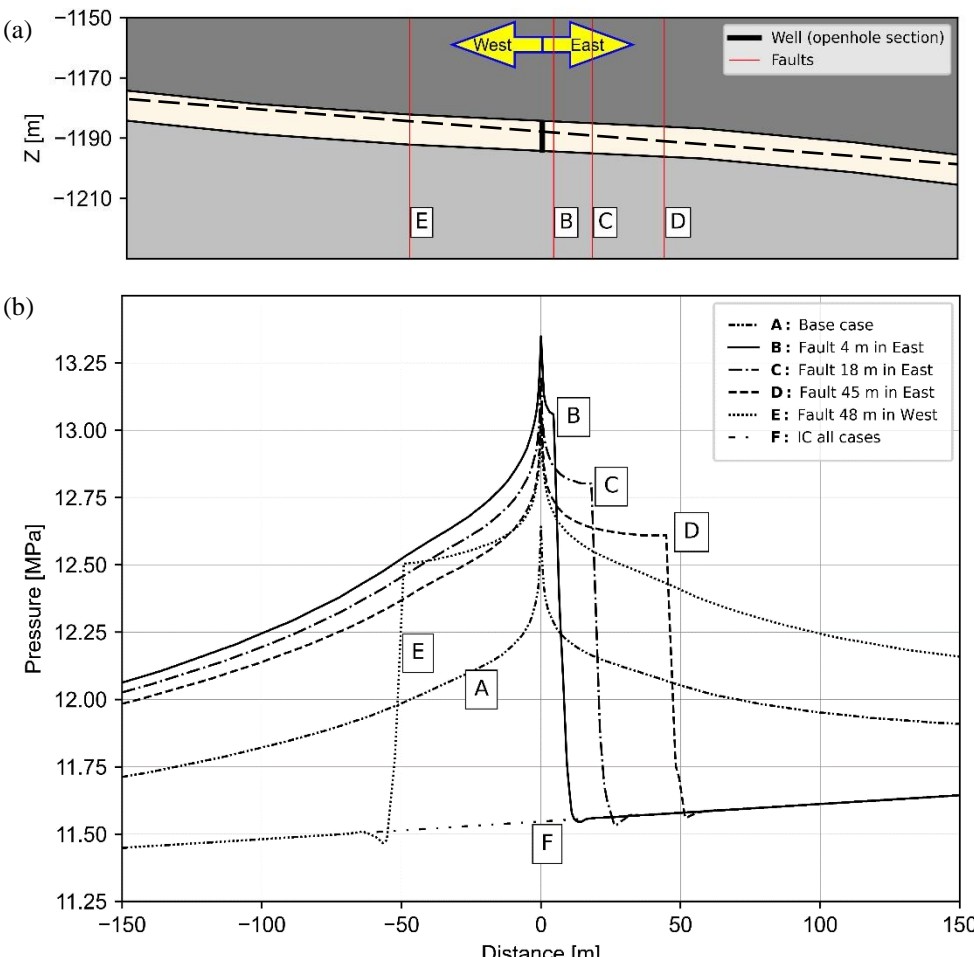

 **Figure 9: Total pressure increase of five simulated cases at the end of the first injection cycle. The cross section in subplot a indicates the position of the traces used for plotting the pressure data of five different scenarios and the initial condition (IC). Negative values for distance represent the western side of the well. To make the curves more readable, scenarios are labelled as A, B, C, D, E, and the initial condition as F.**

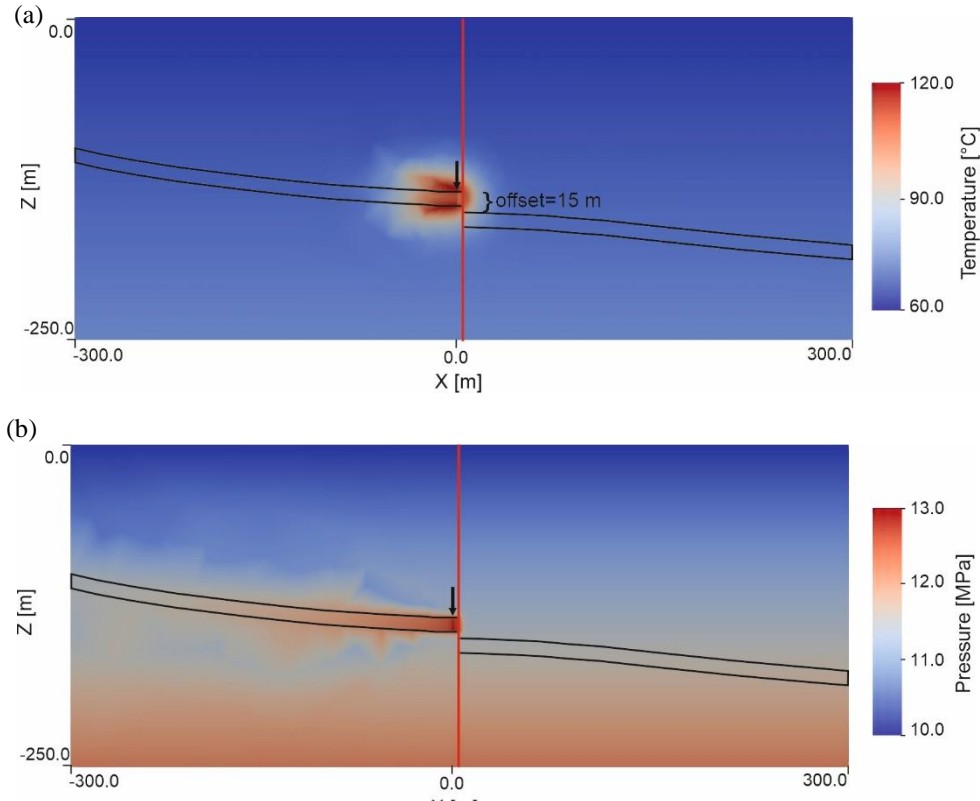

**Figure 10: a) Temperature changes in a cross section of the DeepStor model at the end of the last production cycle. b) Pressure regime in the model after the first injection cycle (6 months). In both subplots location of the well is highlighted by a black arrow in the middle of the model. The fault is represented as a continuous thick red line which locates 4 m in the east of the well and has a fixed 15 m offset. The thick black line also represents the boundaries of the reservoir layer.**

Figure 11 is a contour plot of the total pressure distribution within the reservoir layer. A surface parallel to the tilted reservoir

layer is chosen to create this plot. The trace line shown in Figure 9-a is extended in the y direction to transform it from a line to a surface, making it applicable to the contour plots. In both plots, the well is located in the center with 0.0 and 0.0 coordinates. The first notable point is that pressure is accumulating alongside the contact of the reservoir with the matrix. Instead of spherical pressure plumes, contour lines propose an elliptical high pressure regime with a major axis perpendicular to the fault surface. Despite the negligible difference in the fault distance between Figure 11-a and b, the pressure values are

higher in the case with the fault on eastern side of the well.

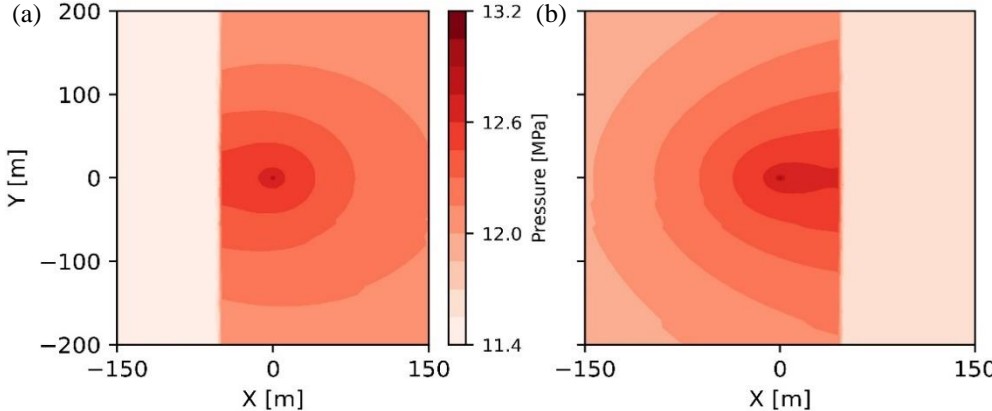

**Figure 11: Total pressure changes after the first injection cycle in two scenarios. The well position is in the center of both plots (coordinates=0.0 and 0.0). The fault location is easily distinguishable by the sharp change in the pressure data: 48 m in the west of well (a) and 45 m in the east (b). Negative and positive values for the x and y axis are relative to the position of the well.**

The presence of the arbitrary fault in the DeepStor model can be identified in the calculated pressure values from the top of the well. Figure 12 shows the history of the total pressure values on the openhole section during the first year of the HT-ATES operation. The location of the fault, either in the east or west, impacts the pressure. The fault distance in the two scenarios is the same (8 m) but in different directions from the well. Due to the pressure accumulation in the downdip direction, a fault with the same distance on the eastern side of the well can increase pressure more than the same one on the 430      western side. The slight difference between the solid black and dashed red curves is detectable in Figure 12.

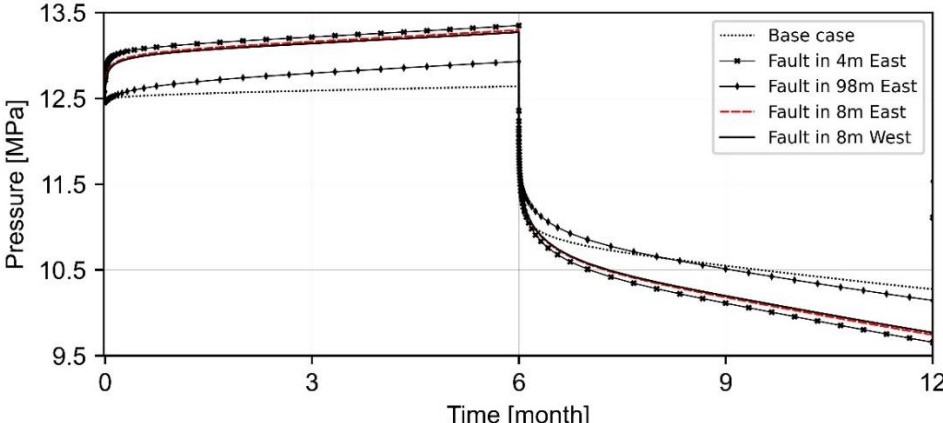

**Figure 12: Total pressure evolution in the well during the first injection and production phase. Higher pressure accumulation in the east of the well can be observed by the slight difference between dashed red (Fault in 8m East) and solid black (Fault in 8m West) curves.**

## 4 Discussion

The developed meshing workflow streamlines the incorporation of geological models and their uncertainties into numerical simulations. This study used generic initial models and introduced arbitrary uncertainties but the same strategy can be applied to real world cases. The discussion section first addresses the existing limitations of the workflow. The included geological uncertainty is later discussed to be applied in both the exploration and development phases.

### 4.1 Limitations of the Workflow

While offering a starting point for automating the meshing process, the developed workflow has limitations. Our current workflow is limited to creating vertical faults, whereas they can be inclined in reality. To investigate the impact of the dip angle, a new scenario with a 67° dipping fault was compared to the existing vertical fault case. The most extreme case with the fault located 4 m east of the well was chosen for this comparison. As Figure 13 shows the temperature distribution and well temperature profiles are identical for both cases confirming the insensitivity of the simulation results to the considered variations in dip angle. This conclusion applies solely to the DeepStor model with its specific configuration.

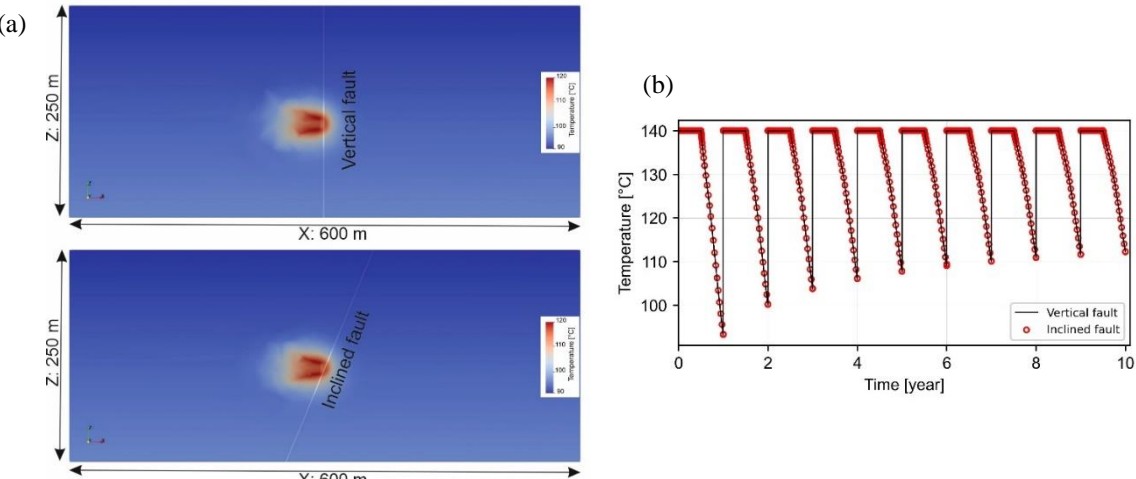

**Figure 13: a) Temperature change after the last production cycle in the DeepStor model with a vertical (top) and inclined (bottom) fault. b) Calculated temperature at the top of well over 10 years of simulation.**

The workflow can include only N-S striking faults. Similar to the special case shown in Figure 13, a fault plane with a 5° deviation has been tested but results remained similar to the case with 0° deviation. This insensitivity to fault dip and orientation is specific to the DeepStor model and in other applications and settings, e.g. tracer flow in a multi-fractured reservoir (Dashti et al., 2023), results can be highly sensitive to them. Another limitation is the workflow's inability to place faults beyond existing grid points (vertices) in the geological model. While increasing resolution can expand available locations, faults still remain confined to predefined locations.

Despite these constraints, the chosen scenarios effectively assessed DeepStor's performance risks. The study revealed negligible impact on thermal performance and a maximum pressure increase of 10 % at the injection well for the closest fault location (4 m). Moreover, increasing the upper distance (112 m) further diminishes the impact of the fault.

While constructing the geological model in advanced tools like Petrel, GemPy, or Leapfrog can provide more scenarios for uncertainty analysis, integrating them with mesh generators is cumbersome. MeshIt (Cacace and Blöcher, 2015), as a mesh generation tool, also aims to address geological models but still requires manual intervention for complex surface creation.

## 4.2 Exploration campaign design

GGB was presented in this study to detect the possible impacts of geometrical uncertainty on the HT-ATES's thermal performance. While all material properties and BCs in our simulations are fixed and derived from the base case of a published document, the geological model, i.e., the mesh, varies. For the chosen parametrization, the heat plume radius even after 10 years of continuous injection and production is still about 40 m around the hot well. Introduced geometrical uncertainty to the GGB case is generic but the proposed workflow is applicable for any real case with its unique complexity/uncertainty. The complex top and bottom surfaces of the reservoir also hardly play any role in the heat distribution of the Malm reservoir. In the case of thinner reservoirs (<20 m) a ± 10 m shift can increase/decrease the volume of the reservoir up to 50 %, but the thermal performance of the Malm reservoir in GGB remained independent of such small scale thickness variations. This fact confirms the unnecessity of complex exploration methods for such specific cases like GGB. Dedicating huge efforts to preliminary steps discourages policymakers from investing in renewable solutions like HT-ATES in settings similar to what has been assumed for GGB in this study. In some cases, existing 2D seismic slices of oilfields can bring enough accuracy to generate reliable forecasts. Computationally affordable geological scenario based analyses of the reservoir can save the time dedicated to exploration.

## 4.3 Field development plan

Based on the presented results for DeepStor, the distribution of both the heat and pressure are tightly linked to the inclination of the thin reservoir. Therefore, incorporating the real geology into the planning process can be a critical factor in optimizing the placement of the second well. As the next step, perturbing the depth, inclination, and thickness of the layer can provide us with a range of possible depths that can be expected during the drilling of the second well.

Within the URG, the majority of hydrocarbons are accumulated thanks to the existence of sealing faults. Therefore, DeepStor can also encounter these structural features. Thermohydraulic simulations revealed that only faults located within distances less than the heat plume radius (45 m) can have negative impacts on storage performance. Considering the size of the heat plume, it is highly unlikely to see any effect from or on the Leopoldshafen or Stutensee faults regarding the thermal performance of the system in a 10 year time frame. The target sand layer is very thin and in the case of thicker formations, the impact of faults can be even less important and observable.

The existing trend in Figure 9 and Figure 12 enables a primary forecast of fault distance (in case of having any) merely based on the recorded well pressures. The pressure difference between day five of injection and the initial condition versus the distance of the fault to the well is used to formulate the forecast. It is assumed that on day five of injection, the initial reservoir condition and injection operation have reached equilibrium. This pressure value can also be measured through a hydraulic test conducted on the well. In the base case of DeepStor, the maximum total pressure reaches from the initial 11.5 MPa to 13.3 MPa, representing a 15 % increase at the end of the first injection cycle. Notably, over half of this increase (11.5 MPa to 12.5 MPa) is observed by day five of the simulation. Figure 14 shows the relation of these two variables where the fault distance from the well versus the pressure increase after five days are plotted. All the 14 black dots represent the scenarios in which the fault is located in the east of the well. For comparison, the case with a fault in 8 m distance in the west of the well is also plotted as a circle to present the pressure accumulation in the downdip direction. To address the worst case scenarios and be as pessimistic as possible, the forecast has been founded only on the base of the faults located on the east of the well. A simple exponential function with three degrees of freedom provides an acceptable level of accuracy (RMSE=0.013 MPa) for the prediction. More simulations can strengthen the presented forecast scheme. Due to the discussed limitations and lack of enough scenarios, we here present the possibility of formulating such simple forecast for a complex reservoir. In the case of generating more simulations, advanced methods like machine learning can also be used. Once developed, other arbitrary distances can be fed into the predictor and the pressure value on day five of injection will be returned without making meshes and running the numerical simulations. As a limitation of our meshing workflow, the fault has been located only at specific distances while the proposed predictor can work for any distance.

After conducting the test phase in reality and measuring the pressure value on day five of injection, the data can be inserted into the predictor to back-calculate the distance of the fault (if present). In the case of finding discrepancies between prior assumptions about the fault location with output of the predictor, the geologic model can be updated. However, the validity of this inversion scheme strongly depends on the accuracy of the chosen modelling assumptions like the material properties used (Table 1) and including only one fault. Otherwise, the difference between measured and calculated pressures can originate from any other sources like petrophysical properties. Global sensitivity analyses shed light on the effect of each parameter on the response of the system. In the case of measuring material properties with error levels less than the sensitive range of the system, the proposed forecast scheme can be more reliable for predicting the underground structural model and performing independently of the parametrization.

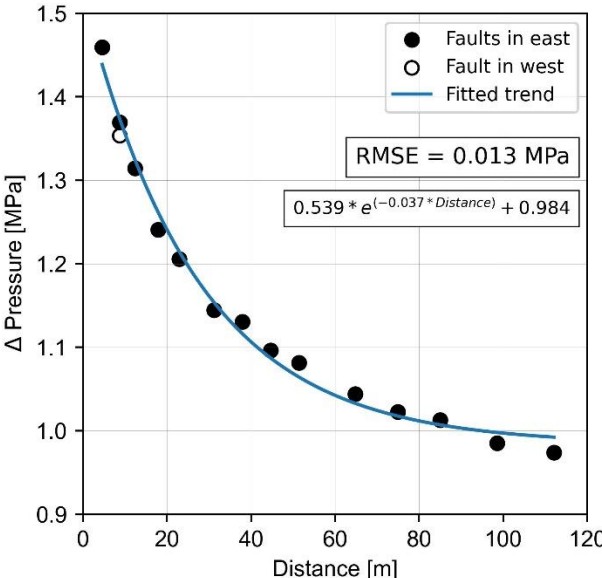

Figure 14: Difference between the well pressure on day five of injection and initial condition (Δ Pressure) versus the distance of the arbitrary fault to the well. The continuous line represents an exponential function with three degrees of freedom.

## 5 Conclusion

In the framework of uncertainty quantification, we have developed a tool applicable to complex geological structures. This study demonstrates a geological scenario based analysis of HT-ATES in two showcases. A new implementation in GMSH provided us with the possibility to automate the generation of complex geological surfaces to overcome the time demanding process. The developed automated workflow in Python brings the possibility to make several meshes composed of surfaces with arbitrary shapes. This workflow also enables a fast generation of finite element meshes using one single block of code in Python without manual adjustments. Generated meshes will link the geological uncertainty of the models to numerical simulators. We used the geological uncertainty as a key input for decision making in different phases (exploration to development) of the HT-ATES.

A HT-ATES is simulated for Geneva as the second most populated city in Switzerland. In GGB, randomly generated geological surfaces are used to assess the sensitivity of results to the geometry of the reservoir rather than the material properties of the model. The GGB model confirms the independence of the temperature from the geometry of the Malm reservoir. The rough structure of the Malm layer can be detected even through 2D seismic slices. Therefore, surveys for finding the exact morphology of the top and bottom surfaces with higher accuracies are unnecessary for such cases. This study highlights the necessity of running computationally affordable simulations before any exploration campaign.

The porous sand layers existing within Meletta beds beneath KIT campus are also promising storage space. For DeepStor adding one more level of complexity (a vertical sub-seismic fault) to interpreted data expresses the performance risks such as

possible significant heat losses and/or pressure increase. With the proposed material properties, the presented evaluation on

DeepStor proved that only in cases where the fault is closer than 45 m to the well, the thermal performance of the system can be negatively affected. The effect on the thermal recovery of the well is hardly observable but the overall dimension of the heat plume can change due to such faults in the vicinity (<45 m). Numerically calculated pressure values at the well location can decipher the faults even in 118 m distances assuming the fixed and certain petrophysical properties. The relation between pressure changes and the location of the introduced fault is used in this study to establish a case specific forecasting

scheme for detecting possible locations of the barriers in the DeepStor model.

The adjacency of the proposed site to oil depleted reservoirs is a big advantage but the real experience of HT-ATES in such locations is still immature, hence first order estimates from risk analyses need to be conducted. Further studies are required to address also the challenges associated with DeepStor including the geochemical interaction or the impact of residual hydrocarbons in the formation. Adding new functionalities to the developed Python script of the DeepStor model can also

enable a more comprehensive uncertainty analysis by perturbing the strike and dipping angle of the sub-seismic fault. Integrating geomodelling tools with mesh generators also offers a promising approach to expand the scope of uncertainty analysis beyond solely varying the fault location, allowing for the inclusion of additional degrees of freedom.

*Code and data availability*. GMSH can be accessed via the published releases on the official GitLab repository at

https://gitlab.onelab.info/gmsh/gmsh. Required data and developed workflows for running the model for both of the showcases are fully documented and available in GitHub (https://github.com/Ali1990dashti/GeoMeshPy/tree/main/Examples/Storage_Models) and Zenodo (https://zenodo.org/records/10256834) repositories of the first author.

*Author contributions.* AD: Conceptualization; Methodology; Simulation; Validation; Code development; Writing– original

draft. JCG: Conceptualization; Geological modelling; Supervision; review & editing. CG: Code development; Writing – review & editing. FB: Geological modelling; TK: Conceptualization; Supervision; Writing – review & editing

*Competing interests*. The authors declare that they have no conflict of interest.

*Acknowledgements.* Ali Dashti is receiving the financial support from The German Academic Exchange Service (Deutscher Akademischer Austauschdienst: DAAD) to do his PhD in Germany as the Research Grants-Doctoral programmes in

Germany 2019/20. This organization is appreciated for giving the opportunity to researchers. The study is also part of the Helmholtz portfolio project Geoenergy. The support from the program "Renewable Energies", under the topic "Geothermal Energy Systems", is gratefully acknowledged. The authors are grateful to Dr. Guido Blöcher, Prof. Guillaume Caumon, and Prof. Florian Wellmann for their insightful reviews and comments that significantly improved the quality of this manuscript. Dr. Mauro Cacace is acknowledged for his fast and efficient editorial handling. Authors appreciate the support of Prof. Eva

Schill (eva.schill@kit.edu) for the data availability and geological model of the DeepStor. Dr. Denise Degen

(denise.degen@cgre.rwth-aachen.de) is appreciated due to her support and constructive comments. Fruitful comments of Kai R. Stricker (kai.stricker@kit.edu) regarding the numerical modelling section are wholeheartedly acknowledged.

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
