# Peer review of "Developing meshing workflows in GMSH v4.11 for Geologic Uncertainty Assessment of the High-Temperature Aquifer Thermal Energy Storage"

_Geoscientific Model Development, 2023_

## Referee Comment (RC1)

[referee-annotated manuscript omitted]

---

## Author Response (AR1)

Dear Reviewer,

Thank you very much for your careful review of the manuscript and helpful comments. Authors appreciate your insights and suggestions, which have helped us to improve the quality of the work. All your comments are addressed below, and a new version of the manuscript with track changes will be uploaded.

*This paper reports on a mesh-based numerical simulation approach to assess the effect of structural uncertainty in geothermal models. The method primarily relies on recent features of the GMSH library, and some THM simulation results are shown on various geological scenarios for two geothermal sites. The problem of mesh updating is challenging and the results are interesting, so I think this paper deserves publication in GMD. However, even though I agree that the proposed workflow has a lot of potential to investigate more advanced settings, the modelling hypotheses remain relatively simple, so I see this paper as closer to a proof of concept than to a conclusive study on the considered geothermal reservoirs, so I think that the conclusions of the sensitivity study should probably be moderated. Therefore, and for the reasons listed below, I recommend major revisions.*

- Thank you very much for your comments regarding our manuscript and for taking the time to review the manuscript.

Main comments

*1. In the GGB case study, the independent depth perturbation of layer hanging wall and footwall and the reference isopach geometry seem like relatively strong modelling assumptions. Could it happen that the reservoir thickness is not constant, e.g., that the overall dip of either interface is subject to uncertainty? What if a larger variogram range was used? Could it happen that some faults be present in the reservoir? Is it consequential to assume constant permeability in the Malm patch reef deposits? Could that change the conclusion? Could you envision to use seismic line interpretations to locally reduce horizon uncertainties around these lines, and possibly use larger uncertainties away from the seismic sections? I don't mean that all of these suggestions should be undertaken for the paper to be published, but I think the wording of the conclusions should be modulated and that the discussion should discuss the uncertainties more thoroughly.*

- The GGB case was designed in this study to evaluate the effect of thickness variations i.e. structural uncertainty on the performance of thick reservoirs. The reservoir is subjected to arbitrary uncertainty because the model is a generic one rather than a real case. We wanted to avoid falling into the realm of unknown unknowns. Our workflow is not uncertainty-dependent and for real cases, any source of realistic errors can be superposed to the structural models. For example, we can define a function for the error that has an inverse relation with the distance from the boreholes. References like Wellmann and Regenauer-Lieb (2012), Wellmann et al. (2010), Wellmann and Caumon (2018), etc. delved into the uncertainty topic. We rather want to quantify the impact of structural uncertainty using two cases. As you mentioned in the annotated version, ± 8 and 10 m are very optimistic for the GGB case. Therefore, we have changed the range to ± 10 and 15 m for the top and bottom contacts of the reservoir, respectively. It is assumed that available well data in the area of the 3D seismic survey allowed for a well-seismic tie. Countless degrees of freedom can be mentioned for the GGB case (and also DeepStor) but we tried to keep the parametrization similar to what exists in the literature. All your comments about the motivations behind our strategy for uncertainty are right. We should mention that in the submitted manuscript we introduced a tool that can be applied to far more complicated cases. The reservoir in the GGB case can also have fractures but such a setting has been out of our scope because then we could not compare our results to what has been done with Collignon et al., 2020. If we vary too many parameters, then it is not possible to control any specific one. Homogeneous permeability across a reservoir is not realistic but for the sake of modelling, we do not consider heterogeneities and give more consideration to the structural uncertainties of the reservoir geometry, i.e. particularly thickness variations of up to 25% at max.
- The thermo-hydraulic influence of faults in thin reservoirs was evaluated for the DeepStor case. There, the reservoir is also slightly tilted which has a minor impact. The wording and structure of the manuscript will be completely modified to present our message more clearly.

*2. In the DeepStor case study, it is great to see the addition of one fault into the model, but could it be that more faults exist? I suspect the conclusions would be quite different if the injector happened to be on a horst or a graben, even more so if there is compartmentalization. As in the previous case, there is also the question of the potential impact of sedimentary heterogeneity. This makes the regression part on pages 18-19 restricted to the relatively simple modelling assumptions. This should be acknowledged.*

- For the DeepStor, in 3D seismic data (not yet allowed to be published) no seismically major fault could be documented within the studied area (1 km × 1km × 250 m). We added one (arbitrary) sub-seismic fault to evaluate the impact on the reservoir's storage performance with varying distances from the well. The heterogeneity of petrophysical properties in the thin reservoir matters as outlined by the sensitivity analysis of Stricker et al. (2020). We used the data of Stricker et al. (2020), which are based on a review of published petrophysical data from sandstones in the Meletta beds across the Upper Rhine Graben, which may best document their heterogeneities. However, we are aware that the data originate from samples taken in different areas and depths, which definitely matters (particularly the poro-perm properties). We are also aware of the fact that samples from boreholes undergo elastic unloading affecting their poro-perm properties. But we need something to start with. Model calibration can be done if hydraulic test data are available. The proposed regression is tightly linked to the parameterization we used in this study. We wanted to highlight relationships between pressure, temperature and the occurrence of possible (sub-seismic) faults as hydraulic boundary conditions for future hydraulic testing and operational data interpretation. This may be also used for other low-thickness heat storage reservoirs. New explanations will be added to the regression part to make the idea behind that regression more clearly and avoid misunderstandings.

*3. Given the small amplitude of perturbation, could a simple mesh displacement be used in the GGB case study instead of a remeshing, as for instance done by (Tertois and Mallet, 2007)?*

- Thanks for this hint. Publications from Jean-Laurent Mallet and his colleagues are pivotal in the realm of meshing geological models. The mesh displacement (based on control points) can be a very useful option but in cases like GGB making two new surfaces in the Python API of GMSH is more robust. In addition, the mesh size has been refined close to the surface and it is computationally more efficient to first create the surface and then refine the mesh instead of moving the refined mesh based on the control points. In the second approach, we may lose the quality of the mesh or not follow the control point correctly. Another point is that we wanted to present this functionality which can be later used for cases with larger uncertainties. Based on your comment we increased the uncertainty range and the meshing process was fast and efficient.

*4. I think it would be very useful to also publish the python scripts for the DeepStor project, as they would show a concrete example about how to use the non-manifold features of GMSH.*

- A fully explained Jupyter notebook is prepared to elaborate on how the mesh for the DeepStor has been generated. The notebook will be uploaded to the Zenodo and GitHub repositories. The only issue is the exact morphology of the reservoir in DeepStor. Again, authors are not yet allowed to publish the detailed structural model of the DeepStor. A thin layer (10 m like the reservoir in DeepStor) will be meshed in the uploaded example. The dimensions of the model in the published code are the same as the DeepStor model and only z values are replaced. The morphology of the presented synthetic model is even more complicated than the reservoir of the DeepStor and a fault also exists in that model.

*5. In its present state, the discussion is mainly a summary of the findings and an extension of the results section (including a regression method to locate the fault based on the sensitivity analysis). I would recommend to move the regression part in the results section, and to improve the discussion section. Overall, structural uncertainty issues have been addressed in other reservoir contexts or with other methods before, so I recommend to*

*integrate some more discussions on the method in the light of previous literature reviewed by (Wellmann and Caumon, 2018), especially : (Holden et al., 2003; Jolley et al., 2007; Seiler et al., 2010; Cherpeau et al., 2012; Irving et al., 2014; Julio et al., 2015; Huang et al., 2015; Patterson et al., 2020; Legentil et al., 2023). The last sentence of the conclusion on chemical processes would also probably better fit in the discussion.*

- In our manuscript, we have not added anything new to the ideas behind structural uncertainties (informative references are mentioned in your comment). In fact, we included them in two heat storage cases. In the first case (GGB), we wanted to document which level of uncertainty we could accept before investing time and money into the refinement of the subsurface data. This is section 4.1 of the discussion (Exploration campaign design). For the DeepStor case, we wanted to see the effect of errors (ignoring the sub-seismic fault) on the system's behavior. As the sub-seismic fault has not been observed, we cannot be sure about its locations, and the regression part of the discussion wants to suggest a predictor for it. We cannot also simulate every possible location of the fault and such predictors can help us to avoid running thousands of simulations. Therefore, we presented our discussion around these aspects. However, to document our awareness of aspects that play a role beyond our modeling we modified the manuscript accordingly.

*Form*

*1. I found the current structure of the manuscript a bit difficult to read. I could help to change the structure by considering first the methodology (current sections 2.3 and 2.4), then one section for each application case. This would make the simulation results closer to the geological settings, and probably avoid going back and forth in the manuscript. For instance, Table 1 gives some geological elements such as formation thicknesses, which would be useful in Section 2.1. I think such reorganization would also help to better separate the generic aspects of the method and the specific choices made for the two considered case studies.*

- Thanks for this hint. We tried to apply your suggestion, but we came back to the original structure. Presenting the information about each case in one section makes that section very long. In section 2 of the manuscript (Uncertainty and Numerical model developments) we first present the structural model and the idea behind the uncertainty of each case. Then the meshing process for both cases is explained and at the end, the details of simulations (like BCs) and material properties are presented. In section 2.1 we will refer to Table 1 for the thicknesses. We will also add more details about meshing for either case which makes the whole section longer.

*2. Some parts of the methodology would call for a bit more details. For instance, did you constrain the geometry of the layer at the injector/ producer location in the GGB case? Do you assume a constant fault displacement in the DeepStor case? (Note that if the displacement is variable, I guess this reinforces the conclusions). What are the lateral boundary conditions in both case studies? Importantly, I don't clearly understand when reading the manuscript how the fault is integrated into gmsh. Providing the python script doing this would be great, but additional details on the management of non-manifold lines and the displacement between the fault and the horizon cutoff lines would be welcome in the text, as this is a challenging problem. Another question is whether this displacement is applied so as to keep the block perforated by the well at the same depth? On the displacement, you may want to have a look at the following references: (Calcagno et al., 2008; Georgsen et al., 2012; Godefroy et al., 2018).*

- All your questions will be addressed in the new version of the manuscript. But some short answers here: in the structural model, it is assumed that in the points where the wells are touching the upper (-1100 m) and lower (-1200 m) contacts of the reservoir, the depth value is a certain data. We assumed a constant fault displacement for the DeepStor case to make the model as pessimistic as possible due to the purpose of simulations which is evaluating the system's reaction in case of embedding a sealing fault in it. In all the 16 faulted scenarios the fault displacement exceeds the reservoir thickness. If we make it less and allow some contact between the reservoir on two sides of the fault, the impact of the fault as a confining barrier decreases. Neither temperature nor pressure BCs are set on the side faces, hence they follow the gradient. All the faces of the models are considered as open to flow. The meshing section for the DeepStor case is completely rewritten. Each step of meshing is elaborated and a new figure is added

to make the meshing process more understandable. In the DeepStor case contact of the well with the top and bottom of the reservoir is fixed and certain.

*3. In the case study presentations, it would be great to summarize the average thickness / poro / perm/ thermal conductivity of the various formations on a stratigraphic column or in a table.*

- Table 1 contains mentioned information about the basement, reservoir and caprock units for both the GGB and DeepStor. The data for both the cases are coming from the literature and a simple table can be more convenient than a stratigraphic column to present the values.

*4. When giving the results timings, please mention the RAM, number of nodes and processors used for HPC computation.*

- The text will be updated.

*5. I don't understand the "time stepping details" (page 8 and 9). Does this refer to the CFL condition for the numerical time step, to the injection schedule, or both?.*

- This part is trying to explain the time discretization. The injection schedule is fixed, for example in GGB we have four months of injecting hot fluid in the hot well and producing from the cold well. Then, we have two months of rest (in which there is no injection and production) followed by four months of production in the hot well and cold fluid injection in the cold well. Then, to have reliable results from the FE solver, we started with very short time steps for each cycle and then increased the size gradually. This way, the solver (MOOSE) can converge more efficiently. In the first cycle (four months of injecting hot fluid in the hot well and producing from the cold well) the time step size increases from one hour to 10 days. Then, the time step size decreases to one hour at the start of the rest cycle (two months) and increases gradually to 20 days at the end of this rest period. At the start of the next four-month cycle, the time step size is forced to be one hour and increases to 10 days at the end.

*6. Some figures could be improved: please show the well on Fig. 3. Fig. 7 would be easier to read by showing semi-transparent or cutoff horizons and fault. Fig. 7 and 8 could be merged, as they are essentially two ways to visualize the same thing. Caption should focus on describing the figure and do not need to stress how the chosen presentation helps to read the figure.*

- Fig. 3 and 7 will be updated. Captions will also be modified. Your handwritten comments on the manuscript are suggesting to merge Fig. 8 and 10. However, in Fig. 8 the pressure value across the reservoir unit is plotted while in Fig. 10 the pressure values are coming from one point (top of the open-hole section where injection and production are happening).

*7. Please mind that the topology for all the GGB scenarios is always the same. Only the geometry changes.*

- Thanks for this comment. The terminology will be corrected accordingly.

*8. More generally, the text has room for improvement in terms of grammar and style. Some sentences need revision or clarification. Some parts of the text read more like a commercial pitch than a scientific paper. Some*

*redundancies could be reduced, especially in the results and discussion sections (e.g., page 13). I highlighted these problems in the attached annotated document.*

- The text will be revised thoroughly. We will remove all the redundancies to make the message of the paper concise, clear, and understandable. Thanks for your hints on the annotated file. Your comments will be considered.

Best regards

Ali Dashti (on behalf of the co-authors)

Dear Reviewer,

Thank you for your careful review of the manuscript and helpful comments. Authors appreciate your insights and suggestions, which have helped us to improve the quality of the work.

*In the manuscript with the title "Developing meshing workflows for Geologic Uncertainty Assessment in High-Temperature Aquifer Thermal Energy Storage", the authors present approach to combine geological modelling, uncertainties, mesh, generation, and geothermal simulations within one workflow. The work contains a combination of previous developments around automated meshing approaches in combination with two dedicated applications to aquifer thermal heat storage (ATES).*

- We appreciate the careful review of the manuscript and constructive comments. All comments (and also the comments from another reviewer) are implemented with track changes in the modified version of the manuscript. In the following also your comments are discussed and replied.

*Even though the work has many important aspects, the overall aim is not entirely clear: a main focus is to test the influence of geological uncertainties on ATES simulations. This was partly evaluated in previous work (if I am not mistaken?) and here mostly extended to the consideration of a fault. The question then also arises if the fault can be detected or not (which is also a question of sensitivity) - and this aspect is then quite similar to the work of Konrad et al., e.g.:*

*Konrad, F., Savvatis, A., Degen, D., Wellmann, F., Einsiedl, F., & Zosseder, K. (2021). Productivity enhancement of geothermal wells through fault zones: Efficient numerical evaluation of a parameter space for the Upper Jurassic aquifer of the North Alpine Foreland Basin. Geothermics, 95, 102119.*

*Degen, D., Veroy, K., Freymark, J., Scheck-Wenderoth, M., Poulet, T., & Wellmann, F. (2021). Global sensitivity analysis to optimize basin-scale conductive model calibration–A case study from the Upper Rhine Graben. Geothermics, 95, 102143.*

*(And maybe also others)*

*A main difference is that, in this work presented here, the (re-)meshing is actually automated - this is different to the approach of Konrad et al, where different scenarios were tested. However, the analysis itself in the manuscript is also only shown for discrete scenarios (e.g. Fig. 8, 10, 11) - and therefore, the advantage of an automated workflow is somewhat diminished (as it could equally be established with a completely manual or semi-automated approach in a reasonable time).*

- As stated by the reviewer (re-)meshing for complex geological models is the main focus of this study. Previous studies on GGB (Collignon et al. (2020)) and DeepStor (Stricker et al. (2020)) mainly focused on the sensitivity analysis in simplified reservoir geometries. For the GGB case we had indeed no limitations to make new scenarios as we can generate unlimited number of scenarios and test them. In the updated notebooks of GGB (https://github.com/Ali1990dashti/GeoMeshPy/tree/main/Examples/Storage_Models/GGB_Case and https://zenodo.org/records/10256834), 101 different geometries are generated and can be meshed. In this study we only presented three of them because the considered range of uncertainty (updated to be ± 10 and ± 15 m for the top and bottom contact of the reservoir, respectively) did not affect the temperature and pressure fields. We argued that this result can be used to design a financially efficient exploration campaign (see text for further details). For DeepStor we have implemented the fault in different parts of the model reservoir according to the varying spacing from the injection well and obtained results for them. A new section is added to the manuscript which provides details implementing the fault in the model. In DeepStor we have some limitations for the location of the fault: we can only locate the fault at places where we have a point grid. In fact, we had to stick to the resolution of our vertices (in x direction) as derived from the geological model. Two notebooks

(uploaded in Zenodo: https://zenodo.org/records/10256834 and Github: https://github.com/Ali1990dashti/GeoMeshPy/tree/main/Examples/Storage_Models) explains how the mesh for the DeepStor case was generated.

*A more significant contribution could be achieved if the geological parameters of the fault (position, dip, fault permeability) were implemented in a real sensitivity analysis. This would also help to clarify the contribution: in the manuscript, it is partly difficult to follow the logic of the work and to see where exactly the novelties lie. If a clear QoI would be defined and, subsequently, a structured SA would be performed (ideally a global SA, see also ref above, which may be difficult due to computational issues), the contribution of the work would be much clearer.*

- The introduced fault is considered not to be detected by 2D-/3D-seismic surveys. We assumed that its strike will be parallel with the major N-S striking faults. Simulations are done in a way to represent the worst case scenarios in which the fault will make a barrier throughout the model. For this purpose, changing the dipping angle does not make a big difference because we want to have offsets higher than the reservoir thickness. This way, the fault surface will make an impermeable boundary for the reservoir layer by juxtaposing it with the impermeable basement. We can superpose more levels of uncertainty to the model but then it cannot possible to track the influence of each one.

*Please also note that the meshing software MeshIt (Cacace & Blöcher) also enables an automation of mesh generation to a certain degree (reasonably fixed topology). Please specify the difference/ advantages of the approach taken here:*

*Cacace, M., & Blöcher, G. (2015). MeshIt—a software for three dimensional volumetric meshing of complex faulted reservoirs. Environmental Earth Sciences, 74, 5191-5209.*

- MeshIt is a great tool for making meshes in complex geometries. However, in MeshIt the user still needs to do some manual work compared to the proposed workflow in this study. For example in the case of GGB we developed a workflow in which the user should only push a point cloud into GMSH and the mesh will be generated automatically without any manual adjustment like selecting the border points and so on. The workflow also leverages functionalities of Python for automating the mesh generation in GMSH. For example, it is required in MeshIT to import each surface as a single input file (as .txt,.csv, etc.) but in Python API of GMSH a single numpy array or a nested list can contain all the surfaces of all the realizations that we need and there is no limitation for the number of surfaces and realizations (all the details are also explained in the notebook uploaded in Zenodo(https://zenodo.org/records/10256834) and Github(https://github.com/Ali1990dashti/GeoMeshPy/tree/main/Examples/Storage_Models)). This way, one single for loop enables making several meshes even without looking into the import, surface generation, meshing, refinement, embedment, physical properties assignment and export processes. The recently uploaded notebook for DeepStor also shows how fast and automated the workflow is for embedding fault planes in the mesh with the least amount of manual adjustments.

*Another option would be a full forward UQ with hundreds of draws from the prior distributions - even if this is mentioned in the paper at some points ("a myriad of watertight FE meshes", "workflow exports as much [sic] as required stochastic geological meshes"), it is not actually shown. Examples which really show the results of automatically generated meshes over a wide range of parameters would help to see the general suitability of the method.*

- Thanks for this suggestion but it has been beyond the focus of this manuscript. This suggestion can be considered as an outlook for us. Such sentences ("a myriad of watertight FE meshes", …) are written to introduce the capabilities of the presented workflow. For the GGB case it did not matter if we change the top and bottom surfaces three times one 100 times. Meshing is the bottleneck between geological

models and numerical simulations and we tried to solve. For the DeepStor also we moved the fault 15 times (due to the resolution of the geological model).

*In any case, a more structured analysis would not only help to highlight the contribution of the work (and the advantage of using an automated meshing methods), but also clarify the structure of the paper and increase the readability.*

- Thanks for this comments. The whole structure of the manuscript is revised and rewritten to clarify our message more clearly.

*Very important to note is that, in this manuscript, only geometric parameters are considered - and the rock properties remain fixed. Requires explanation or, at least, clarification from the beginning onwards - and also in the discussion and conclusion!*

- It is added to the new version.

*Please find below some more comments to specific sections in the text. The manuscript also contains several grammatically wrong, incomplete, and partly unclear sentences. Some of these are mentioned below, but these are by far not all of them - a thorough revision wrt grammar and logic is suggested.*

- The whole text is revised and corrected.

*49-50: Please provide more reasoning or a reference for this claim.*

- The text is corrected.

*55: A (conceptual) sketch about ATES systems, the structural and petrophysical elements, and their respective uncertainties would be helpful at this point.*

- We have added a new figures about how the mesh has been generated for GGB and DeepStor. Petrophysical elements are coming from the literature and we mentioned them in a table. Fig. 1 also shows the reservoir geometry for one of the scenarios from the GGB case.

*91: Randomised noise: what exactly does this mean? Are (vertical position) uncertainties added to the nodes, which are then sampled IID (seems so, considering line 95)? Very important in a geological/ geophysical context: spatial correlation! The variations are by no means independent. Please explain in more detail how exactly the surfaces are modified and which rational is behind choosing or not choosing spatial correlations. Note that uncorrelated noise will typically lead to lower effects of the uncertainty!*

- The GGB case is generic and we followed the parametrization and geometry proposed by Collignon et al. (2020). In our model we assumed that 3D seismic data is giving a grid of points representing a surface and also two boreholes drilled into the reservoir were seismic well tie can be done. We started from a box with flat planes representing the top and bottom of the reservoir and we are certain about four points which are the contact of two boreholes with the reservoir. We can define a function in which the noise has a direct relation with the distance from the boreholes. This way, we may generate a

reservoir with concave or convex surfaces. In our study we wanted to show the level of the irregularity that can be meshed. As Fig. 1 show the mesh for GGB contains a lot of small hills and depressions. If we correlate the noise, there would be a relation between them and making the surface can be easier than what we have generated via the random noise. We have increased the level of uncertainty to be $\pm$ 10 and $\pm$ 15 m for the top and bottom contact of the reservoir, respectively and results did not change. The text is updated based on this comment.

*93: Points are just points - what makes a surface is the connection between them, i.e. the topology.*

- Thanks for this hint. It is added to the manuscript.

*133: See comments above: why only so few scenarios? If the process is automated: why no distributions and random samples? Choosing scenarios is adding significant bias to the analysis!*

- The process is automated but still limitations exist. For example the vertical fault should be positioned at locations that a point grid exist there and we had to deal with the resolution of the available model (~14 m in x direction and ~26 m in y direction). The whole fault embedment is elaborated in the uploaded notebook and a lot details are also added to the updated version of the manuscripts. Regarding the scenarios for DeepStor we were biased to choose the worst case bases. Therefore, we put the fault in as much as possible locations in the eastern side of the well.

*158: Adding functionality for more complex topologies is certainly a key aspect of the approach! So, do you mean to say that, in this work, the complexity is extended through the consideration of a fault? Please be more specific on the level of complexity that can be achieved: would it also work for fault networks with multiple intersections, pinching out of layers, finite faults?*

- In the previous study (Dashti et al. (2023)) a geological model with very simple surfaces were tested but at the moment the surface fitting method is so robust that can be used for any type of geometries. It can work for fault networks but the relation between the networks should be identified beforehand. An example in Github (https://github.com/Ali1990dashti/GeoMeshPy/tree/main/Examples/No_Fault) can explain how to transfer from generated fault networks in GemPy to correctly discretized meshes in GMSH. Dashti et al. (2023) published a Python library (GeoMeshPy) to link GemPy with GMSH which works for some types of the models. GemPy can generated a wide variety of geological models (with onlap relation, pinch out, etc.) which cannot be handled by GeoMeshPy and GMSH. But in case of adjusting the input into readable and correct format for GMSH, it can be done.

*191: Mesh sensitivity: results?*

- Results are not presented to keep the manuscript short. Otherwise, two extra figure for mesh sensitivity of each case can be provided.

*196: General-purpose PDE environment*

- Is added.

*199: of TIGER*

- Is corrected.

*200: Is this functionality (mixed meshes) actually used?*

- It is used. In our meshes boreholes are represented by 1D lines and fault planes by 2D surfaces. Then, we can add another dimension in the input file to it. For example a thickness will be added to the fault plane to make it a volume.

*204: Repetition*

- Is corrected.

*Sec. 2.4 in general: as simulations take a prominent part, more information about physics required in the form of employed equations and coupling.*

- New explanation about TIGER are added. We could add the governing equations that TIGER uses to solve fluid-flow and heat-transfer but all of the details are already mentioned in publications like Gholami Korzani et al., 2020.

*223: Repetition*

- Is corrected.

*257: Unclear sentence*

- Is corrected.

*Fig. 4 and 5: no difference visible, figures in this form not helpful. Suggestions: 2-D instead of 3-D (third dimension does not help in Flg 4 but rather distract); show a single example and maybe a difference plot, if there are any differences visible?*

- Fig. 4 is presenting also a 3D view into three different reservoir geometries. With the 3D figure the reader can have a better view on the different reservoir geometries. As mentioned there is no difference in the heat plume in Fig. 4 hence we presented a line plot to check for any difference between three scenarios and nothing is noticed. We used this observation in our discussion to argue about the decision making for the required levels of accuracy in our data.

*Fig. 7: Same consideration*

- This figure is updates. As another reviewer commented, we added the cutoff of the fault plane to it. In Fig. 7 the 3D view and also cross section in middle of the model are observable. Dimension of the heat plum in y direction is also completely different for the scenario with a fault in 4 m in the east of the well.

*Fig. 8: best figure in the paper (as clearly some aspects are visible and marked). Numerical question: what is the reason for the negative "bump" just before the increase (and at other locations) in    E? Please explain, comment on numerical validity of result.*

- The negative "bump" is a negligible mesh effect. We tried to solve it via refining the mesh on the fault plane but it was adding a huge amount of computational cost without so much improve. The fault in the model was considered as a big plane and refinement on it can add a lot of more elements. The pressure values go back normal just meters before and after the fault plane. We refined the mesh on the fault plane and also boreholes up to levels that results stabilized. Regarding the validity of the results, unfortunately we have not any measured data to calibrate the model with. However, the results especially in Fig. 8 make sense. We can see that everything is happening between the borehole and the fault plane. For example in the scenario E, the eastern side of the fault stays in the initial conditions or for B, C and D the western side of the faults is unperturbed.

*336: Uncertainties - into what?*

- Is corrected, into numerical simulations.

*340: Automation mentioned - but not really used (and shown) in the paper - see comments above. "As much": "as many". How many? What is the reasoning behind the choice here?*

- The text is corrected to avoid misunderstandings.

*342: "Any type of data"? Please explain!*

- Is corrected and explanations are added.

*349: Usually, it is not of interests to broaden the uncertainty. I guess you mean to say: extend the consideration of uncertain parameters/ consider more aspects of uncertainty?*

- Thanks for this hint, the text is corrected.

*352: Topological uncertainty: really considered? Seems to me that topology (at least of the geology) remains fixed. Please review concepts on topology.*

- Thanks for this hint, the text is corrected.

*361: Statement would only be true in the range of the assumptions taken here.*

- The text is corrected.

*368: A bit confusing: you write above that uncertainties about structures do not matter - here they do. The difference is due to the two examples - but here the important point: a more clearly structured SA would help!*

- The text is corrected to be clearer. For the GGB case the uncertainty did not matter so much (of course this conclusion is specific to the parametrization we have chosen from the literature) whereas for the DeepStor the real geological model in which the reservoir was inclined and the sub-seismic fault mattered. Stricker et al. (2020) presented a detailed SA for the DeepStor. Collignon et al. (2020) also presented the same for GGB. Here we chose to the base case parametrization and modify the structural model.

*385-390: Why this analysis? Also Fig. 11: not clear why results are relevant. Please explain/ expand.*

- This part of the discussion want to say that we can use pressure data to get some ideas about the geological model. Fig. 11 shows the correlation between pressure data and the distance of the sealing fault. Such strategy allows us to rebuild or at least revise the structural model after getting some measure data. Of course the assumption is that we are certain (!) about the material properties. Typically the geological models are in the simulations and parameters are tuned in a way to match the measurements but in this manuscript we want to see what happens if we change the geological model. New explanations are added to present the message in a more clear way.

*417: "a myriad": see comments above.*

- The text is corrected.

*423: Unclear sentence.*

- The text is corrected.

Based on the comments we replied here and also comments from the first reviewer, the text and structure of the manuscript is revised thoroughly.

Best regards

Ali Dashti (on behalf of the co-authors)

---

## Referee Report (RR1)

The current manuscript is about an automated workflow between structural geological models and physics-based numerical models for evaluating structural uncertainties in HT-ATES. This workflow is tested on 2 examples: an example with variable reservoir thickness and an example of a wedging reservoir with a sealing fault zone. Of course, an automated workflow is of great importance for mesh-based simulations, as re-meshing can be avoided. Nevertheless, the applications should be chosen so that they are not just a proof of concept but can also be used for more sophisticated uncertainty analyses. In the applications presented, a variation in the thickness and position of a sealing vertical fault zone within a thinning geological layer was evaluated. This is a good start but currently seems to be the limit of the method used. Furthermore, only 3 cases were calculated for the first reservoir and only 17 cases for the 2nd reservoir, which is the lower limit for an uncertainty analysis. The content of the manuscript is adequate for GMD, but major corrections are required in order to accept the manuscript.

P1L19: What does "thick" reservoir mean. There is a noise function on the top and bottom surfaces which alters the reservoir thickness and it should be described that way.

P1L23: The uncertainty analysis was carried out over a range of 4 m to 118 m for the distance from the fault zone to the well. What is the basis for this range. There is no information whether this comes from the thermal radius of the storage cycles or from geological modeling.

Abstract: It is not clear, what is the motivation and the scientific question that is to be answered? It appears that the main consideration is the thickness variation due to some random noise functions and the distance of a sealing fault zone from the wellbore. It seems that the presented approach is limited here for a vertical fault zone with an offset greater than the reservoir thickness. Only in this way the fault zone could be implemented as a hydraulic barrier. What natural scenario is this assumption based on? Does this approach work for inclined faults with less offset and acting as barrier or pre-dominant flow-path, too?

P2L35: ATES characterization using push-pull tests are described in: "Best practices for characterization of High Temperature-Aquifer Thermal Energy Storage (HT-ATES) potential using well tests in Berlin (Germany) as an example, Geothermics, Volume 116, 2024, 102830, ISSN 0375-6505, https://doi.org/10.1016/j.geothermics.2023.102830."

P2L45: (e.g., well configuration, transmissivity, flow rate, conductivity, …) → (e.g., well configuration, transmissivity, flow rate, and conductivity)

P3L64: "…transfers stochastic structures from geological uncertainty models to a fast and reliable numerical meshing tool…". No geological uncertainty model was described or presented in the present study. How should the scientific community evaluate whether the transformation of a geological model into a numerical model is possible using the presented approach? Here a new mesh is generated and not an existing model is transferred.

P3L85: "…flow rates of <0.5 l/s…". Flowrates should be related the pressure responses. It is not clear if the provided value a design parameter or a limitation by the reservoir performance or submersible pump?

P4L97: "To perturb the geological model, a randomized noise is superimposed on the top and bottom surfaces of the reservoir layer." But it is not clear which conceptional geological model is responsible for such a noise function. What would be the geological process behind?

P4L100: "For the bottom surface, the range of perturbation is increased to ± 15 m due to the decrease in the quality of seismic data with depth." Again, what is the basis or measurement for assuming that magnitude and distribution of noise? It seems to be a random number.

P5L127: A normal fault with a vertical offset of more than the reservoir thickness is presented. Main question is, what is the stress state to generate a normal faulting with such an offset? Generally, normal faults dip with 40 to 70 degree.

P6L145: The sealing fault is represented by an offset exceeding the thickness of the aquifer. To mimic the sealing fault this could even be done by truncating the model at a designated distance. I believe it is worth to check if a truncated model (at the distance of the sealing fault) would provide the same result as the model with fault offset. This could be a discussion point.

P6L149: It is still unclear if the fault itself is sealing or the offset greater than the thickness of the reservoir generates the sealing feature. I assume that the second case is the one discussed/represented in the study. Otherwise the question arises, how a sealing fault is implemented in the FEM. Is the permeability of the fault set to zero? Is flow perpendicular to the fault possible (fault is transparent to flow)?

P6L151. Ones more, the normal fault is represented as vertical fault. Is this a limitation of the method? How dip and dip-direction are quantified. If no quantification was made, dip and dip-direction should be analyzed in the uncertainty study.

P6L152-155: As mentioned before only few scenarios were considered for this uncertainty analyses. One advantage of the automatized workflow should be the performance. Why not thousands of simulations were performed for varying distances, strike directions and dip angles?

P11L253: Moose is updated frequently. Therefore, TIGER should be maintained. By my knowledge this is not the case any longer. Therefore, a future use of TIGER and the reproducibility of the presented simulations maybe questionable? Maybe the perspective or alternatives like GOLEM (Cacace, M. and Jacquey, A. B.: Flexible parallel implicit modelling of coupled thermal–hydraulic–mechanical processes in fractured rocks, Solid Earth, 8, 921-941, https://doi.org/10.5194/se-8-921-2017, 2017.) should be mentioned as well.

P12L282: How you deactivate the temperature BC for production scenarios? Temperature BC should be assigned for injection mode. For production mode these BC should be deactivated. Was this deactivation considered, if so it should be described shortly.

P13L310: If no boundaries are defined in general a no flow boundary is the default assignment. I wonder if Tiger would deal differently. Therefore, I suggest to check if the lateral borders are open to flow or no-flow boundaries.

Figure6: What is the reason for an injection temperature of 39C and not of 40C as the reservoir temperature?

P15L341: "Despite the negligible difference, the case with a fault located 48 m in the west of the well has the best performance…". Yes, but it seems the in a distance of 45 m and more the fault has no influence anymore. It seems that the thermal radius/plume has a radius of approximately 45 m. An explanation is missing why a fault zone having a distance of more than 45 m should influence the simulation results.

Figure 7: What is the vertical offset of the fault for these scenarios. It is mentioned to be more than the aquifer thickness. Why we do not see a sub-figure of such a simulation. I suggest to add a figure showing at least one example of the reservoir with fault offset and the related pressure and temperature field. In figure 8 the fault is visible but not the offset. Maybe this figure can be improved be showing all geometric features.

P16L355: Ones more, what is the thermal radius of your base case. It can be approximated by using the presented estimation in: Daniel T. Birdsell, Benjamin M. Adams, Martin O. Saar, Minimum transmissivity and optimal well spacing and flow rate for high-temperature aquifer thermal energy storage, Applied Energy, Volume 289, 2021, 116658, ISSN 0306-2619, https://doi.org/10.1016/j.apenergy.2021.116658.

It seems to be about 50 m. Therefore, a fault in 48 m distance has no influence?

Figure10: As ask before, what would be the effect of truncated model domain at a distance equivalent to the simulated sealing fault. I assume you will obtain similar results without simulating the fault offset.

P21L411: The chapter "4 Discussion" reads like a summary and not like a discussion. I suggest to rewrite this paragraph to introduce to the subsequent discussion point.

P21L419: This study was not based on geological models. Therefore, this sentence seems to overestimate the potential of the presented approach which is an automated mesh generation and should be either modified or deleted.

Figure12: Such diagrams are known from well test analysis and should be compared to other analytical and numerical solutions for the case of "one no-flow boundary" as presented for example in: http://oilproduction.net/files/Fekete_WellTestApplications.pdf

---

## Author Response (AR2)

Dear Reviewer,

Thank you very much for your careful review of the manuscript and helpful comments. Authors appreciate your insights and suggestions, which have helped us to improve the quality of the work. All the comments are addressed below and a new version of the manuscript with track changes is uploaded.

*The current manuscript is about an automated workflow between structural geological models and physics-based numerical models for evaluating structural uncertainties in HT-ATES. This workflow is tested on 2 examples: an example with variable reservoir thickness and an example of a wedging reservoir with a sealing fault zone. Of course, an automated workflow is of great importance for mesh-based simulations, as re-meshing can be avoided. Nevertheless, the applications should be chosen so that they are not just a proof of concept but can also be used for more sophisticated uncertainty analyses. In the applications presented, a variation in the thickness and position of a sealing vertical fault zone within a thinning geological layer was evaluated. This is a good start but currently seems to be the limit of the method used. Furthermore, only 3 cases were calculated for the first reservoir and only 17 cases for the 2nd reservoir, which is the lower limit for an uncertainty analysis. The content of the manuscript is adequate for GMD, but major corrections are required in order to accept the manuscript.*

**General Reply**

The remarks were on two principal aspects:
A) Where are the limitations of the method,
B) There are too few models to perform an uncertainty analysis
The manuscript has now been updated based on the comments of the reviewer. First, we detail these major remarks in a rather general reply:

A) The methodology developed has some inherent limitations. As such, a structural variation can only be performed on existing grid points, i.e. vertices (Figure 4) and the fault has to be vertical with a fixed N-S strike (Figure 4). Otherwise, our method offers a high flexibility, on the definition of the surfaces of the aquifer (morphology, dip, and variable thickness) as well as on the location of the (vertical) fault. Applying this method for thermohydraulic calculations offers a high variation of simulations.

It may be noted that the impact of a lower dip angle is rather negligible for layers with defined thickness (e.g. even a 45° dip angle would increase only the apparent near field transmissivity by a factor of 1.4, but the far field transmissivity remains unaffected). The following cartoon illustrates this simplification:

[Figure]

For the sake of comparison, two new special fault models including a fault of 67° dipping angle or ± 5° strike angle have been elaborated and tested against the existing vertical scenario. The comparison was made on the example of the vertical fault in Chapter 3.2 (i.e. fault located 4 m in the east of the well). The following figures show the results in two cases that look almost the same:

[Figure]

Comparison of vertical to 67° inclined fault: figures (left: contour plot of temperature after 10 years i.e. the last production cycle, right: well temperature over time) show a nearly perfect agreement of both cases, i.e. vertical versus inclined fault.

[Figure]

Comparison of vertical to ±5° deviated fault: figures (left: contour plot of temperature after 10 years i.e. the last production cycle, right: well temperature over time) show a nearly perfect agreement of both cases, i.e. N 0° versus N 5° fault.

Corresponding clarifications regarding the limitations of the method are added to Chapter 2.2 of the manuscript.

B) The following provides an example of the unlimited application of the code regarding the number of scenarios for the Greater Geneva Basin (GGB). The final resulting storage capacity (temperature production) does not vary and was therefore not presented in the manuscript. For further illustration, we uploaded examples of 101 surfaces ready to be meshed and - due

to file storage restrictions - accordingly 14 meshes in Zenodo[1] as well as in Github[2]. Easily, more examples could also be calculated using the uploaded scripts.

In the manuscript, we generated the arbitrary (generic) geometries by adapting the top and bottom surfaces of the reservoir. Out of 11 simulations, three had been presented in the manuscript. As discussed the reservoir performance is independent of the considered uncertainty on that scale. This result is later used in Chapter 4.1 to design an efficient exploration campaign for cases similar to GGB. The following illustrates this statement for 11 scenarios on behalf of the temperature field (after 10 years, i.e. the last production cycle) in the Malm reservoir. The different reservoir geometries are visible by their surface topographies, but they do not have any impact on the temperature field:

[Figure]

[1] https://zenodo.org/records/10256834
[2] https://github.com/Ali1990dashti/GeoMeshPy/tree/main/Examples/Storage_Models/GGB_Case

[Figure]

[Figure]

Corresponding clarifications are added to Chapter 3.1 of the manuscript.

**Detailed reply to individual questions**

*P1L19: What does "thick" reservoir mean. There is a noise function on the top and bottom surfaces which alters the reservoir thickness and it should be described that way.*

> The original term "thick" was ambiguous and has been clarified in the revised manuscript. In the GGB case, the reservoir has a varying thickness, which was initially set to be of uniform 100 m thickness. Our methodology applied a thickness variation of up to 25 %. In the DeepStor case, the reservoir remained uniform with a 10 m thickness.
> The text in the abstract is changed to be:
> "Developed meshing workflow is applied to two case studies: 1) Greater Geneva Basin with the Upper Jurassic ("Malm") limestone reservoir and 2) the 5° eastward tilted DeepStor sandstone reservoir in the Upper Rhine Graben with a uniform thickness of 10 m. In the Greater Geneva Basin example, the top and bottom surfaces of the reservoir are randomly varied ± 10 m and ± 15 m, generating a total variation of up to 25 % from the initially considered 100 m reservoir thickness."

*P1L23: The uncertainty analysis was carried out over a range of 4 m to 118 m for the distance from the fault zone to the well. What is the basis for this range. There is no information whether this comes from the thermal radius of the storage cycles or from geological modeling.*

> The 4 m to 112 m range is chosen to evaluate the effect of the fault on the heat distribution and also examine the possible relation between the location of the fault and pressure response at the well's location. These points are added to the manuscript. Limitations for the location of the fault are also added to Chapter 2.2 of the manuscript.

*Abstract: It is not clear, what is the motivation and the scientific question that is to be answered. It appears that the main consideration is the thickness variation due to some random noise functions and the distance of a sealing fault zone from the wellbore. It seems that the presented approach is limited here for a vertical fault zone with an offset greater than the reservoir thickness. Only in this way the fault zone could be implemented as a hydraulic barrier. What natural scenario is this assumption based on? Does this approach work for inclined faults with less offset and acting as barrier or pre-dominant flow-path, too?*

> The abstract is re-written. Limitations of the method are also addressed in the new version of the manuscript.

*P2L35: ATES characterization using push-pull tests are described in: "Best practices for characterization of High Temperature-Aquifer Thermal Energy Storage (HT-ATES) potential using well tests in Berlin (Germany) as an example, Geothermics, Volume 116, 2024, 102830, ISSN 0375-6505, https://doi.org/10.1016/j.geothermics.2023.102830."*

This recently published work is added to the text.

*P2L45: (e.g., well configuration, transmissivity, flow rate, conductivity, …) → (e.g., well configuration, transmissivity, flow rate, and conductivity)*

The text is updated.

*P3L64: "…transfers stochastic structures from geological uncertainty models to a fast and reliable numerical meshing tool…". No geological uncertainty model was described or presented in the present study. How should the scientific community evaluate whether the transformation of a geological model into a numerical model is possible using the presented approach? Here a new mesh is generated and not an existing model is transferred.*

The text is updated to be:
"This study expands the application presented in Dashti et al. (2023) by introducing an automated workflow that generates meshes for complex structural models, enabling the quantification of relevant processes in HT-ATES."

*P3L85: "…flow rates of <0.5 l/s…". Flowrates should be related the pressure responses. It is not clear if the provided value a design parameter or a limitation by the reservoir performance or submersible pump?*

This value comes from the literature, i.e. Guglielmetti et al. (2022), highlighting the low transmissivity of the reservoir in that specific location.
A new sentence is added to the text:
"The flow rate has also been low due to the reservoir's characteristics in that specific location." Guglielmetti, L., Heidinger, M., Eichinger, F., and Moscariello, A.: Hydrochemical Characterization of Groundwaters' Fluid Flow through the Upper Mesozoic Carbonate Geothermal Reservoirs in the Geneva Basin: An Evolution more than 15,000 Years Long, Energies, 15, 3497, https://doi.org/10.3390/en15103497, 2022.

*P4L97: "To perturb the geological model, a randomized noise is superimposed on the top and bottom surfaces of the reservoir layer." But it is not clear which conceptional geological model is responsible for such a noise function. What would be the geological process behind?*

The presented work primarily aims to demonstrate that the developed script can generate various complex meshes for the GGB while the source of uncertainty remains generic. Collignon et al. (2020) investigated a storage case for the Malm reservoir with a uniform 100 m thickness represented as a box with flat surfaces. In contrast, our study modifies the top and bottom surfaces of the reservoir in a way to make complex and more realistic geometries for the reservoir. The developed workflow for this instance remains entirely independent of uncertainty. Any type of noise (error) can be superimposed on the initial model with flat top

and bottom surfaces. The 2D section in Figure 1-a presents the concept of superposed generic uncertainty. Chapter 2.1 has also been modified accordingly.

Collignon, M., Klemetsdal, Ø. S., Møyner, O., Alcanié, M., Rinaldi, A. P., Nilsen, H., and Lupi, M.: Evaluating thermal losses and storage capacity in high-temperature aquifer thermal energy storage (HT-ATES) systems with well operating limits: insights from a study-case in the Greater Geneva Basin, Switzerland, Geothermics, 85, 101773, https://doi.org/10.1016/j.geothermics.2019.101773, 2020.

*P4L100: "For the bottom surface, the range of perturbation is increased to ± 15 m due to the decrease in the quality of seismic data with depth." Again, what is the basis or measurement for assuming that magnitude and distribution of noise? It seems to be a random number.*

Publications like Stamm et al. (2019) and Lüschen et al. (2011) have been considered to choose these two ranges. Chapter 2.1 is modified in a way to present the reasoning behind the noise values more clearly.
Stamm, Fabian Antonio, Miguel de la Varga, and Florian Wellmann. "Actors, actions, and uncertainties: optimizing decision-making based on 3-D structural geological models." Solid Earth 10.6 (2019): 2015-2043.
Lüschen, E., Dussel, M., Thomas, R., & Schulz, R. (2011). 3D seismic survey for geothermal exploration at Unterhaching, Munich, Germany. First Break, 29(1).

*P5L127: A normal fault with a vertical offset of more than the reservoir thickness is presented. Main question is, what is the stress state to generate a normal faulting with such an offset? Generally, normal faults dip with 40 to 70 degree.*

(See general comments above)
Another noteworthy point: when altering the dipping angle, a fault could be detected in a nearby well in close proximity to the fault.

*P6L145: The sealing fault is represented by an offset exceeding the thickness of the aquifer. To mimic the sealing fault this could even be done by truncating the model at a designated distance. I believe it is worth to check if a truncated model (at the distance of the sealing fault) would provide the same result as the model with fault offset. This could be a discussion point.*

Thank you for bringing up this interesting point. Truncating the model can lead to higher temperature values within the model. As shown in Figure 8-b, heat diffusion occurs through the fault plane, potentially influencing the overall temperature distribution. Additionally, truncating the model may introduce boundary effects that could impact the thermal and hydraulic performance of the system during injection/production operations. We investigated this approach for one of the cases and examined the performance difference between a model with a fault located 4 m east of the well and a truncated model. As evident in the following figure, the truncated model exhibits a higher efficiency compared to the faulted model.

[Figure]

Therefore, we did not change the modelling approach on this point.

*P6L149: It is still unclear if the fault itself is sealing or the offset greater than the thickness of the reservoir generates the sealing feature. I assume that the second case is the one discussed/represented in the study. Otherwise the question arises, how a sealing fault is implemented in the FEM. Is the permeability of the fault set to zero? Is flow perpendicular to the fault possible (fault is transparent to flow)?*

The offset is creating a sealing. The text is updated:
"This pessimistic assumption enables the prediction of the worst case scenarios for the storage in which a sealing fault completely blocks the reservoir by juxtaposing the reservoir against impermeable matrix."

*P6L151. Ones more, the normal fault is represented as vertical fault. Is this a limitation of the method? How dip and dip-direction are quantified. If no quantification was made, dip and dip-direction should be analyzed in the uncertainty study.*

(See general comments above).
A clarification was added to the manuscript in Chapter 2.2 to explicitly address the limitations of the proposed method:
"The grid resolution in the x direction (14 m in the available DeepStor model) determines the fault location. The workflow is designed to incorporate only N-S striking vertical faults that pass through existing grid points. This is the first limitation of the developed method. For this study, only the barrier effect is relevant and minor changes in the strike direction will not impact the numerical results. Another limitation is the dip angle of the arbitrary fault. For simplicity, the developed script includes a vertical normal fault. For the DeepStor reservoir with a uniform 10 m thickness a change in dip of the sealing fault will have a negligible impact on the simulation results. Even a 45° dip angle would increase only the apparent near field transmissivity by a factor of 1.4, while the far field transmissivity remains unaffected. Additionally, a vertical fault cannot be detected by a planned vertical well, i.e. the well trajectory may intersect the inclined fault."

This point has been added as an outlook to the manuscript:

"Adding new functionalities to the developed Python script can also enable a more comprehensive uncertainty analysis by perturbing the strike and dipping angle of the sub-seismic fault."

*P6L152-155: As mentioned before only few scenarios were considered for this uncertainty analyses. One advantage of the automatized workflow should be the performance. Why not thousands of simulations were performed for varying distances, strike directions and dip angles?*

Figure 4 and its accompanying text in Chapter 2.2 provide detailed explanations of how we incorporated the fault into the base case, displaced the reservoir layer, and split the top and bottom surfaces of the reservoir. Regarding the fault location, we were constrained by the grid resolution in the x direction (14 meters in the existing DeepStor model), which determined the locations where the fault could be positioned. The updated text accompanying Figure 4 addresses these limitations. Additionally, the maximum distance was determined based on the thermal and hydraulic radii. Beyond 118 m, the fault has minimal impact on the hydraulic response of the system at the well location.

*P11L253: Moose is updated frequently. Therefore, TIGER should be maintained. By my knowledge this is not the case any longer. Therefore, future use of TIGER and the reproducibility of the presented simulations maybe questionable? Maybe the perspective or alternatives like GOLEM (Cacace, M. and Jacquey, A. B.: Flexible parallel implicit modelling of coupled thermal–hydraulic–mechanical processes in fractured rocks, Solid Earth, 8, 921-941, https://doi.org/10.5194/se-8-921-2017, 2017.) should be mentioned as well.*

The simulations can be reproduced by GOLEM or available modules of MOOSE like Porous Flow.
The text is updated to make this point clear:
"To reproduce the results, other MOOSE based applications like GOLEM (Cacace and Jacquey, 2017) or available modules of MOOSE, e.g. Porous Flow (Wilkins et al., 2021), can be used."

*P12L282: How do you deactivate the temperature BC for production scenarios? Temperature BC should be assigned for injection mode. For production mode these BC should be deactivated. Was this deactivation considered, if so it should be described shortly.*

A control system is implemented to switch the temperature BC. We can define a set of time intervals: from 0 to 6 months the system should be in the injection mode and from 6 to 12 months in production. Then, a temperature BC will be defined for the target nodes. Now, the switch will look into the time interval and in case of being in the first part (injection phase), the temperature BC will activate, otherwise (6 to 12 months) it won't.
The updated text will have this new information in Chapter 2.4:
"The MOOSE control system dynamically updates the temperature boundary condition (BC) during the simulation. In the injection phase, the temperature BC is applied to the corresponding nodes in the model, either set to 90 °C or 39 °C. During the production phase, the temperature BC is deactivated."

*P13L310: If no boundaries are defined in general a no flow boundary is the default assignment. I wonder if Tiger would deal differently. Therefore, I suggest to check if the lateral borders are open to flow or no-flow boundaries.*

Thanks for this hint. Unfortunately, the wrong terminology has been chosen. It is corrected now. No flow BC which acts like Neumann BC with 0 value have been used. The text is updated: "No flow BCs are considered for side faces of the models."

*Figure6: What is the reason for an injection temperature of 39C and not of 40C as the reservoir temperature?*

For this case the work of Collignon et al. (2020) is followed.

*P15L341: "Despite the negligible difference, the case with a fault located 48 m in the west of the well has the best performance…". Yes, but it seems the in a distance of 45 m and more the fault has no influence anymore. It seems that the thermal radius/plume has a radius of approximately 45 m. An explanation is missing why a fault zone having a distance of more than 45 m should influence the simulation results.*

The reservoir layer also dips eastward and the heat accumulates in the western part of the tilted layer. Therefore, a barrier in the west can increase the heat-trapping efficiency of the reservoir. Of course, in the long term it is a negative aspect because it reduces the available space.
The new explanation is added to Chapter 3.2:
"For the best recovery, the reason is linked to the total volume of the reservoir and upward movement of the low density hot fluid. The reservoir is tilted and hot fluid moves to the updip direction due to the density effect. Then, a barrier in the updip (west) side of the reservoir can block the movement of the hot fluid and make a more efficient heat storage reservoir."

*Figure 7: What is the vertical offset of the fault for these scenarios. It is mentioned to be more than the aquifer thickness. Why we do not see a sub-figure of such a simulation. I suggest to add a figure showing at least one example of the reservoir with fault offset and the related pressure and temperature field. In figure 8 the fault is visible but not the offset. Maybe this figure can be improved be showing all geometric features.*

The vertical offset is 15 m. It is mentioned more explicitly in the updated manuscript. It was impossible to visualize the offset in Figure 8. Hence, a new figure (as Figure 10) is added in the Chapter 3.2:

[Figure]

**Figure 1: a) Temperature changes in a cross section of the DeepStor model at the end of the last production cycle (10 years). b) Pressure regime in the model after the first injection cycle (6 months). In both subplots location of the well is highlighted by a black arrow in the middle of the model. The fault is represented as a continuous thick red line which locates 4 m in the east of the well and has a fixed 15 m offset. The thick black line also represents the boundaries of the reservoir layer.**

*P16L355: Ones more, what is the thermal radius of your base case. It can be approximated by using the presented estimation in: Daniel T. Birdsell, Benjamin M. Adams, Martin O. Saar, Minimum transmissivity and optimal well spacing and flow rate for high-temperature aquifer thermal energy storage, Applied Energy, Volume 289, 2021, 116658, ISSN 0306-2619, https://doi.org/10.1016/j.apenergy.2021.116658.*
*It seems to be about 50 m. Therefore, a fault in 48 m distance has no influence?*

For the DeepStor the thermal radius is about 45 m.

*Figure10: As ask before, what would be the effect of truncated model domain at a distance equivalent to the simulated sealing fault. I assume you will obtain similar results without simulating the fault offset.*

This issue is fully addressed with a figure in another reply.

*P21L411: The chapter "4 Discussion" reads like a summary and not like a discussion. I suggest to rewrite this paragraph to introduce to the subsequent discussion point.*

The discussion presents the applications of uncertainty analysis, which are:
1. Designing an exploration campaign,

2. Correlating the geological parameters with reservoir performance
The text is rewritten to make these points more clear.

*P21L419: This study was not based on geological models. Therefore, this sentence seems to overestimate the potential of the presented approach which is an automated mesh generation and should be either modified or deleted.*

The text is updated to be:
"Geological models and their uncertainty should be transferred into reservoir simulations."

*Figure12: Such diagrams are known from well test analysis and should be compared to other analytical and numerical solutions for the case of "one no-flow boundary" as presented for example in:*
*http://oilproduction.net/files/Fekete_WellTestApplications.pdf*

The diagram establishes a relation between fault's distances with the pressure increase at the well location. Pressure values are calculated numerically with no flow BC.

---

## Author Response (AR3)

Dear topic editor,

Thank you for your careful judgment of the manuscript and helpful comments. Authors appreciate your insights and suggestions. Comments are addressed below and a new version of the manuscript with track changes is uploaded.

*Dear authors,*

*After receiving the comments from the second round of reviews to your manuscript, and based on your responses to the reviewer's criticism, I'm now in a position to provide my comments as topical editor to your submission. I agree with the reviewers' main criticisms (also with respect to the first round of revision) on not well described limitations of the model described in the study. This is especially the case if one considered that the approach has only limited applicability in terms of more realistic geological scenarios. To be limited to prescribed simple fault geometry of uniform (vertical) dip is indeed an important limitations of the procedure. As a matter of facts, fault surfaces show rarely a uniform geometry, but showcase rather varying spatial geometric features (apart from inherent self-affine corrugations). In this respect, I personally found the answers provided to justify the model limitations not scientifically sound, stating that varying the dip of the fault has not direct influence of the thermo-hydraulic response of the latter is against a more than 30 years of literature research. Similarly, I found that the lack of a clear discussion (at least) on a more quantitative uncertainty analysis also comes as an important limitation to the scientific merit of the study. What is the reason of automatizing an approach if not to enable proper sensitivity analysis to be carried out, where the range of effective parameters (geometry and properties) can be quantified in a robust statistical sense? Another important limitation is related to the fault behaviour (likely inherent from the dynamic modelling approach adopted throughout the study), which only provide to test permeable faults if not under the unrealistic assumption of large offset (remember that fault offset is not only a geometric features but comes with important consequences on the fault hydraulic behaviour). This said, I would be willing to consider your submission after another round of major revision, where all these important limitations are discussed in details (also by providing at least hints for future development to overcome the latter.*

The remarks on principal aspects (bold red titles) are addressed in the following:

**1. Limitations of the presented work and possible solutions**

The updated manuscript emphasizes the limitations of the presented methodology further. The developed script for manipulating the DeepStor model has limitations and is not intended to be as robust as advanced geomodelling tools like Petrel, Leapfrog, and Gempy. It was designed specifically to embed one vertical fault with a uniform offset into the base case model of the DeepStor and return features of the geological model as adjusted inputs for the mesh generator. More sophisticated assumptions can be directly integrated into the abovementioned tools and further discretized.

Section 2.2 is updated to detail the technical limitations of the developed workflow. Section 4 (Discussion) has been updated to address the existing limitations and potential solutions in a new sub-section: **4.1 Limitations of the workflow**. The last paragraph of the conclusion also addresses new outlooks for the future direction (based on the existing limitations).

**2. Oversimplified fault surface**

We include a hypothetical fault and look for the potential impact of it and there is a lack of any clue about the fault surface. This way, we cannot put any complex surface topography on the fault plane. Section 2.2 is updated to make this point clear.

**3. *Impact of the fault on the thermo-hydraulic response of the model**

We agree that the dip of the fault typically has a thermo-hydraulic impact. However, in a model with specifications like the DeepStor, the dipping angle did not affect the results. A dip variation of even 25° (at 4 m distance of the injector) on a 10 m thick reservoir did not impact the simulation results over 10 years. This point has been made clear in Section 4.1 of the updated manuscript.

**4. *A more quantitative uncertainty analysis**
- GGB case: 11 models were tested and the uncertainty did not impact results.
- DeepStor case: the location of the fault controls the pressure distribution in the model. Fig. 14 in the updated manuscript presents the relation between the fault location and pressure. More simulations will confirm this general trend. The manuscript shows that a correlation exist between a geological structure (like the fault location) and the model uncertainty. As it was shown in this case, more simulations will not change the uncertainty significantly. Section 4.3 is updated accordingly.

**5. *Fault behaviour and representation in the model**

In our meshing procedure, faults (as 2D planes) are integrated only for displacing the 3D elements. They do not have any significance for the MOOSE simulation and can be considered as being only a virtual plane. Section 2.4 is updated to address this point.